# OVID: OPEN-VOCABULARY INTRUSION DETECTION

**Fujun Han**[1,2]**, Jingqi Ye**[1,3]**, Chenglong Zhang**[2,3]**, Peng Ye**[1,4*]
[1] Shanghai Artificial Intelligence Laboratory
[2] School of Data Science, The Chinese University of Hong Kong, Shenzhen
[3] University of Science and Technology of China, [4] The Chinese University of Hong Kong
hanfujun@cuhk.edu.cn

## ABSTRACT

Various vision intrusion detection models have achieved great success in many scenarios, *e.g.*, autonomous driving, intelligent monitoring and security. However, their reliance on pre-defined classes limits their applicability in open-world intrusion detection scenarios. To remedy these, we introduce the *Open-Vocabulary Intrusion Detection* (OVID) project for the first time. Specifically, we first develop a novel dataset named Cityintrusion-OpenV for OVID, with more diverse intrusion categories and corresponding text prompts. Then, we design a multi-modal, multi-task, and end-to-end open-vocabulary intrusion detection framework named OVIDNet. It achieves open-world intrusion detection via aligning visual features with language embeddings. Further, two simple yet effective strategies are proposed to improve the generalization and performance of this specific task: (1) A **M**ulti-**D**istributed **N**oise **M**ixing strategy is introduced to enhance the location information of unknown and unseen categories. (2) A **D**ynamic **M**emory-**G**ated module is designed to capture the contextual information under complex scenarios. Finally, comprehensive experiments and comparisons are conducted on multiple dominant datasets, *e.g.*, COCO, Cityscape, Foggy-Cityscape, and Cityintrusion-OpenV. Besides, we also evaluate the universal applicability of our model in real scenarios. The results show that our method can outperform other classic and promising methods, and reach strong performance even under task-specific transfer and zero-shot settings, demonstrating its high practicality.

## 1 INTRODUCTION

Vision-based intrusion detection tasks have numerous applications in life, *e.g.*, security, intelligent monitoring, and autonomous driving (Ye et al., 2024; Han & Ye, 2025; Li et al., 2025b;a). Intrusion detection aims to determine whether potential objects go into a specific restricted Area-of-Interest (AoI) (Sun et al., 2020; Shi et al., 2022). Based on whether the camera is moving or not, intrusion detection tasks can be divided into static and dynamic intrusion detection. Static intrusion detection is relatively easy due to the fixed AoI and achieves great progress by some promising strategies, *e.g.*, Histogram of Oriented Gradients (HOG) (Zhang et al., 2015), Conditional Random Field (CRF) (Matern et al., 2013), Adaptive Background Subtraction (ABS) (Stauffer & Grimson, 2000). However, static intrusion detection can not meet the requirements of *real-time* and *accuracy* under dynamic scenes. Fortunately, with the continuous development of computer vision, some promising detection and segmentation frameworks are proposed (Wang et al., 2023; Chen et al., 2018), which provide new schemes and paradigms for solving the problem of dynamic-view intrusion detection, *i.e.*, based on *overlapping pixel points* between objects and AoI (Sun et al., 2020; Shi et al., 2022). Nevertheless, these proposed models can only detect a single intrusion category, *i.e.*, **Pedestrians**. Considering the lack of sufficient practicality, MF-ID (Han et al., 2024b) and MMID-bench (Han et al., 2024c) propose the first concept and task of multi-category and multi-domains intrusion detection, which successfully extends the intrusion categories and domains, *i.e.*, **1→4**. Meanwhile, Ada-iD (Han et al., 2024a) proposes a new active domain adaptation intrusion detection method to further improve the performance of intrusion detection in adverse environments. Although these promising works extend the intrusion categories and scenarios by effective strategies, *e.g.*, Unsuper-

---

[*]Corresponding Author

vised Domain Adaptation, Active Domain Adaptation, their reliance on pre-defined object classes limits their applicability in **Open-World** intrusion detection scenarios, as shown in Figure 1. A pre-defined intrusion detection framework can only detect specific or labeled categories in training sets and can not detect categories unseen and undefined, which severely limits the practicality of intrusion detection. For some unseen intrusion categories, *e.g.*, Car or Truck, the previous models can not give correct intrusion detection results. To address this problem, we propose the **O**pen **V**ocabulary **I**ntrusion **D**etection, namely **OVID**, to detect unseen and undefined intrusion categories effectively.

To accomplish the above OVID task, the greatest difficulty is that there is still a lack of relevant datasets. Currently, some promising datasets, *e.g.*, ImageNet (Deng et al., 2009), COCO (Lin et al., 2014) and specific autonomous driving datasets, *e.g.*, Cityscapes (Cordts et al., 2016b), BDD100K (Yu et al., 2020) are proposed. These datasets are not suitable for our intrusion detection task due to the lack of intrusion labels. Fortunately, in recent works, some encouraging intrusion detection datasets have been proposed, *e.g.*, Cityintrusion (Sun et al., 2020), Cityintrusion-Multicategory (Han et al., 2024b), and Multi-Domain Multi-Category Datasets (Han et al., 2024c). These datasets provide multi-category, multi-domain and intrusion labels ('**N**'/'**Y**'), *i.e.*, '**N**' and '**Y**'denotes No-intrusion and Intrusion. Although these datasets contain multiple categories and domains, they can not

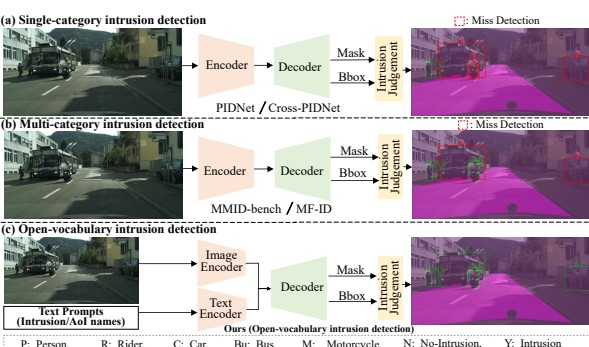

Figure 1: **Workflow comparisons** of different intrusion detection methods. Here, (a), (b), and (c) denote the Single-category, Multi-category, and proposed Open-vocabulary intrusion detection paradigms, respectively. '**?**' denotes the missed detection (False Negative). We can find that previous works can only detect the pre-defined intrusion category; our framework can detect more categories correctly, which demonstrates the validity of our paradigm.

meet the requirements of the OVID task. On one hand, these datasets still lack lots of common intrusion categories, *e.g.*, Car, Bus, Truck. On the other hand, these datasets solely contain image labels without matching text labels, which impairs their practicality in open-world intrusion detection where the model is required to generalize to new, unseen objects. To this end, we propose an extensive and comprehensive intrusion detection dataset, Cityintrusion-OpenV, for the OVID task.

The second difficulty is that there is still a lack of an effective and efficient open-vocabulary intrusion detection framework. Although some promising multi-task intrusion detection frameworks (Sun et al., 2020; Han et al., 2024b) and open-vocabulary detectors are proposed (Kim et al., 2023; Yao et al., 2023), these detectors still can not meet the requirements of the OVID task. The main reason is that the former is constrained by limited detection categories, and the latter can only perform a detection task. Inspired by promising works (Han et al., 2024b), we propose an *effective*, *multimodal*, *multi-task*, and *end-to-end* open vocabulary intrusion detection framework, OVIDNet, to accomplish the task. The input of OVIDNet contains two different modalities, images and text. Subsequently, these inputs are sent to encoders to extract features, then decoded and predicted using the decoder. In the decoder, two simple yet effective strategies are designed to boost the performance of OVIDNet: 1) A Multi-Distributed Noise Mixing strategy is introduced to enhance the location information of unknown and unseen categories. 2) A Dynamic Memory-Gated module is designed to capture the contextual information in complex scenarios. Finally, the intrusion detection results are determined jointly by the upper and lower branches.

In summary, our contributions are as follows: **(1) Novel task and dataset.** To the best of our knowledge, the task of dynamic-view **O**pen **V**ocabulary **I**ntrusion **D**etection is proposed for the first time. This is the first multi-modal try in the vision-based intrusion task. A new benchmark, including a dataset called Cityintrusion-OpenV, and some strong baselines, is given for this task. **(2) Effective design and strategy.** An effective, multi-modal, multi-task, and end-to-end framework, OVIDNet, is designed as a strong baseline for this new benchmark. Besides, two effective strategies are proposed to improve the generalization and performance of the framework, including the Multi-Distributed Noise Mixing Strategy and the Dynamic Memory-Gated module. **(3) Adequate exper-**

**iments and strong results.** Comprehensive experiments and comparisons are conducted to verify the effectiveness of the proposed framework and methods. The results show that our framework not only reaches the current SOTA level but also maintains strong high practicality with task-specific transfer and zero-shot prediction abilities.

## 2 RELATED WORKS

**Traditional Vision-based Intrusion Detection.** Intrusion detection can be divided into static and dynamic intrusion detection. Static intrusion detection has been explored in detail due to its simplicity, *e.g.*, Adaptive Background Subtraction (Stauffer & Grimson, 2000), Histogram of Oriented Gradients (Zhang et al., 2015). However, static intrusion detection does not meet the requirements of dynamic intrusion detection. The primary reason is that images captured by cameras can change at any time, which imposes higher real-time and accuracy requirements. Fortunately, with the rapid development of computer vision, some promising works are proposed to solve the dynamic-view intrusion detection, *e.g.*, PIDNet (Sun et al., 2020), Cross-PIDNet (Shi et al., 2022). However, the practice of these works is limited due to the **single category** of intrusion. To compensate for the limitations, some encouraging works are designed to border the intrusion category, *e.g.*, MF-ID (Han et al., 2024b), MMID-bench (Han et al., 2024c). Although intrusion detection has expanded significantly, the intrusion category remains fixed and single, which seriously hinders the practicality in the real world. To address these, we propose an open vocabulary intrusion detection task (OVID) for the first time, to solve the problem with limited intrusion categories and improve its practicality.

**Open-Vocabulary Detection.** Open-vocabulary detection (**OVD**) aims to detect unseen classes in the training stage in a zero-shot manner. Some promising open vocabulary detectors are designed to solve the OVD task, *e.g.*, Grounding DINO (Liu et al., 2024), YOLO-world (Cheng et al., 2024), and achieve excellent performance on real-world detection tasks. However, these OVD works are not suitable for our intrusion detection task. A major reason is that our OVID task is a multi-task with detection and segmentation simultaneously, not only a single detection task. OpenSeeD (Zhang et al., 2023) proposes a new framework for joint detection and segmentation, but this framework cannot meet the requirements of our OVID task due to a lack of intrusion judgment capability. For this reason, we propose a new framework, OVIDNet, to meet the needs of our OVID task.

## 3 SYSTEMATIC DATASETS

To compensate for the lack of richness in the categories of intrusion detection datasets, we develop an Open-Vocabulary Intrusion Detection dataset, namely **Cityintrusion-OpenV**, for the first time. The detailed automatic generation method of the proposed datasets is presented in **Appendix A.1**. Our new dataset is established on the promising Cityscape (Cordts et al., 2016b). Inspired by promising work MMID-bench (Han et al., 2024c), our new dataset also contains multi-categories and multi-domains. Differently, proposed datasets have more intrusion categories, not 4 categories in Multi-Domain Multi-Category datasets, but all potential/possible **8** intrusion categories in the

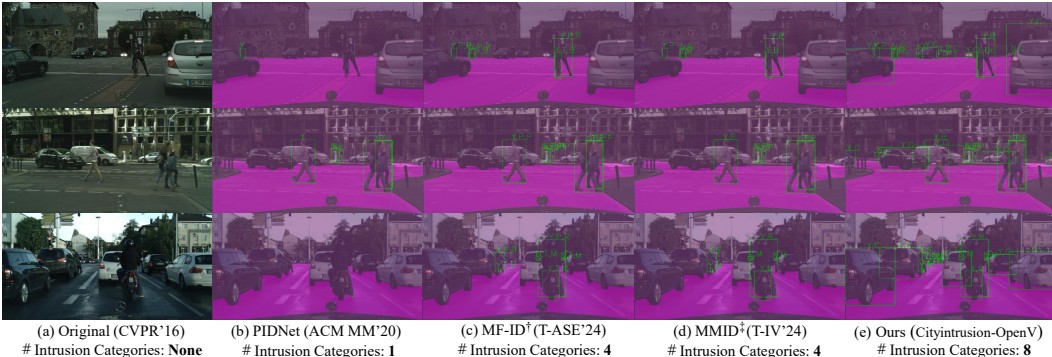

| (a) Original (CVPR'16) | (b) PIDNet (ACM MM'20) | (c) MF-ID[†](T-ASE'24) | (d) MMID[‡](T-IV'24) | (e) Ours (Cityintrusion-OpenV) |
| # Intrusion Categories: **None** | # Intrusion Categories: **1** | # Intrusion Categories: **4** | # Intrusion Categories: **4** | # Intrusion Categories: **8** |

Figure 2: The visualization comparison between our datasets and other promising intrusion detection datasets. [†] and [‡] denote fine-grained and multiple domains, respectively.

cityscape dataset. Following some promising works (Han et al., 2024c), the detail Intrusion('**Y**')/No-intrusion('**N**') labels are provided by Automated Label Processes. And the judgment threshold is also set to **20**. To demonstrate the superiority of our dataset, we first present some visualization comparison, as shown in Figure 2. We can observe that our datasets provide the correct intrusion and no-intrusion labels and have richer categories of intrusions.

Then, we compare the quantitative results between the proposed Cityintrusion-OpenV and other promising intrusion detection datasets, as shown in Table 1. We can find that our dataset contains more intrusion categories and has more sufficient 'Y'/'N' cases per image (**18.03** cases per image in the whole dataset), which significantly improves the intrusion de-

Table 1: The comparison between previous intrusion detection datasets and our datasets. [†] denotes multiple domains.

| Intrusion Detection Dataset Names | Categories | 'Y'/'N' Cases | Cases per image |
|---|---|---|---|
| Cityintrusion (Sun et al., 2020; Shi et al., 2022) | 1 | 4599/15084 | 7.3 |
| Cityintrusion-Multicategory (Han et al., 2024b) | 4 | 5431/22683 | 9.59 |
| Multi-Domain Multi-Category (Han et al., 2024c;a) | 4 | 5431/22683[†] | 9.59 |
| Ours (Cityintrusion-OpenV) | 8 | 24750/37899 | **18.03** |

tection dataset richness ( about **2×** up compared to others). More detailed information, data statistics, and more visualization comparisons are presented in **Appendix A**.

## 4 THE PROPOSED FRAMEWORK AND METHODS

### 4.1 PRELIMINARY

In the proposed OVID task, given a specific object detection dataset $\mathbf{D}^d$, $\mathbf{D}^d = \left\{ \left( \mathbf{I}_i^d, \mathbf{M}_i^d \right) \right\}_{i=1}^{|\mathbf{D}^d|}$ and segmentation dataset $\mathbf{D}^s$, $\mathbf{D}^s = \{(\mathbf{I}_i^s, \mathbf{M}_i^s)\}_{i=1}^{|\mathbf{D}^s|}$, where $\mathbf{I}_i^d$ and $\mathbf{I}_i^s$ denotes the detection and segmentation samples, $\mathbf{M}_i^d$, $\mathbf{M}_i^s$ denotes the corresponding labels. For $\mathbf{M}_i^d$, we usually use four bounding box labels (b) and a category label ($c^d$) to express it, $b \in \mathbb{R}^4$, $c^d \in \mathcal{C}^d$. $\mathcal{C}^d$ denotes the category space of detection dataset ($\mathbf{D}^d$). For $\mathbf{M}_i^s$, we usually use fine labeling (assigning a category $c^s$ to each pixel in the sample $\mathbf{I}_i^s$), $c^s \in \mathcal{C}^s$, $\mathcal{C}^s$ denotes the category space of segmentation dataset ($\mathbf{D}^s$). Besides, we divide the detection and segmentation dataset into $(\mathbf{D}_T^d, \mathbf{D}_T^s)$, $(\mathbf{D}_V^d, \mathbf{D}_V^s)$. $\mathbf{D}_T^d$, $\mathbf{D}_V^d$ denotes the training and validation of detection dataset. $\mathbf{D}_T^s$, $\mathbf{D}_V^s$ denotes the training and validation of segmentation dataset. Following the Open-Vocabulary detection and segmentation paradigms, we can express the training and validation category as $\mathbf{C}_T^d$ and $\mathbf{C}_V^d$, where $\mathbf{C}_T^d$ is base categories and $\mathbf{C}_T^d \in \mathbf{C}_V^d$. The category space of the new categories in $\mathbf{C}_V^d$ are named as $\mathcal{C}^N$, and $\mathcal{C}^N = \mathbf{C}_V^d \setminus \mathbf{C}_T^d \neq \varnothing$. The same is true for the segmentation dataset. For $\mathbf{I}_i$, we use the text Encoder ($\mathbf{E}^\mathbf{T}$) to convert text information into text embeddings, *i.e.*, $\mathbf{F}^\mathbf{T} = \mathbf{E}^\mathbf{T}(\mathbf{Text})$ and use the image Encoder ($\mathbf{E}^\mathbf{I}$) to extra the image feature, *i.e.*, $\mathbf{F}^\mathbf{I} = \mathbf{E}^\mathbf{I}(\mathbf{Img})$. All features are sent to a Decoder ($\mathbf{D}$) for decoding. Finally, the segmentation AoI and bonding box are extracted to calculate the intrusion results and express it as

$$\mathbf{I}_s = \mathbf{J} \left\{ \mathbf{D} \left( \mathbf{F}^\mathbf{T}, \mathbf{F}^\mathbf{I} \right) \xrightarrow{\mathbf{e}} \left( \mathbf{Box}^\mathrm{p}, \mathbf{Aoi}^\mathrm{p} \right) \right\}, \tag{1}$$

where $\mathbf{I}_s$ denotes the final intrusion results. $\mathbf{J}$ denotes the intrusion judgment module. $\xrightarrow{e}$ denotes the extract two key information from the Decoder ($\mathbf{D}$). $\mathbf{Box}^\mathrm{p}$, $\mathbf{Aoi}^\mathrm{p}$ denote the prediction of the bounding box and AoI, respectively.

### 4.2 OVERALL FRAMEWORK

In this section, we introduce the proposed OVIDNet, as shown in Figure 3. We improve the original OpenSeeD (Zhang et al., 2023) to make it more suitable for our OVID task. Firstly, we use two different encoders, text and image encoders, to extract features for text prompts and images. The features are sent to the decoder. In the decoder, we design two effect methods to improve the generalization of the proposed framework: Multi-Distributed Noise Mixing and Dynamic Memory-Gated Module, respectively. Finally, we extract detection results and the segmentation results in the decoder to calculate the overlapping pixel values and determine whether it constitutes an intrusion. Once the overlapping pixel value exceeds a certain threshold ($t$), a warning ('Y/N, Class') will be added to the detected intruder surround box. Note that 'Y', 'N', and 'Class' denote the Intrusion, Non-intrusion, and Class name. (Here, class abbreviations are used instead of complete labels to easily show our results. The detailed correspondence between the two is shown in **Appendix A.4**.)

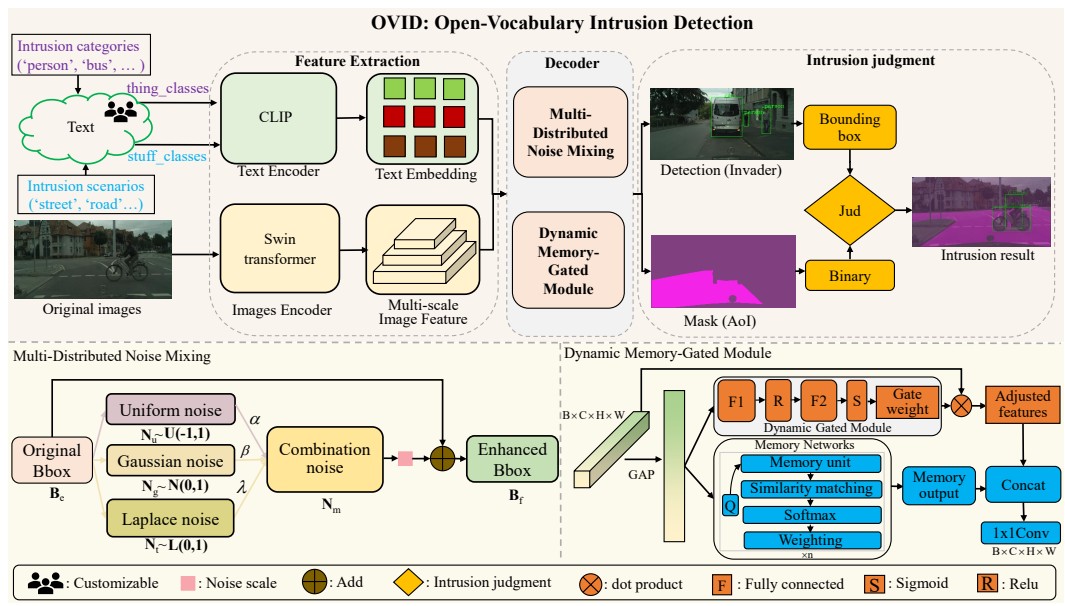

Figure 3: The overall framework and pipeline of our proposed OVIDNet. The input of OVIDNet consists of two different modalities: Text and Images. The text includes some customizable and common intrusion categories and scenarios. The image denotes the corresponding original images. Then, the text and images are sent to different encoders to extract features, *i.e.*, clip and tiny-swin-transformer, respectively. These features will be sent to the decoder for decoding and prediction. In the encoder, we design a multi-distributed noise mixing strategy and a dynamic memory-gated module to enhance generalization in open scenarios. Finally, we extract the predicted bounding box and predicted AoI mask to calculate the overlapping pixels and give the final intrusion results. Once the overlapping pixels are greater than the threshold ($t$), it will be judged as an intrusion. Otherwise, it will be judged as no-intrusion. We use abbreviations to represent the full name to better express the intrusion results. The detailed correspondence can be found in **Appendix A.4**.

## 4.3 MULTI-DISTRIBUTED NOISE MIXING STRATEGY

In the original OpenSeeD model, noise generation methods usually use a uniform noise distribution and a fixed percentage of noise dynamics. Therefore, we can express it as

$$\mathbf{B}_f = \mathcal{C}\left\{\mathbf{B}_e + \mathbf{N}_r \odot \Delta \odot \Upsilon, \ \mathbf{0}, \ \mathbf{1}\right\}, \tag{2}$$

where $\mathbf{B}_e$ denotes the set of the bounding box, $\mathbf{B}_e$ can be expressed by center point $(\mathbf{x}, \mathbf{y})$ and width, height $(\mathbf{w}, \mathbf{h})$. $\Delta$ denotes the range of the disturbance and $\Delta = \left\{\frac{\mathbf{w}}{2}, \frac{\mathbf{h}}{2}, \mathbf{w}, \mathbf{h}\right\}$. $\Upsilon$ is a constant noise scaling factor. $\mathbf{N}_r$ denotes the random distribution and $\mathbf{N}_r \sim \mathcal{U}(-1, 1)$. $\odot$ denotes the element-wise product. $\mathbf{C}$ denotes that all value is clamped between $\mathbf{0}$ and $\mathbf{1}$. However, in the real world, this method can not adapt to dynamic environments and scenarios, *e.g.*, different sizes and changing objects, and challenging intrusion scenarios. Therefore, to address this issue, we propose a new **M**ulti-**D**istributed **N**oise **M**ixing Strategy, as shown in Equation 3. The idea of the proposed Multi-Distributed Noise Mixing Strategy is very simple yet effective. When confronted with complex dynamic environments, models need to cope with the variations of different targets and scenarios. Specifically, for tiny objects, fine-grained perturbations are used to preserve their detailed features. Meanwhile, large-scale perturbations to strengthen the global features for large objects.

$$\mathbf{B}_f = \mathcal{C}\left\{\mathbf{B}_e + (\alpha \cdot \mathbf{N}_u + \beta \cdot \mathbf{N}_g + \gamma \cdot \mathbf{N}_t) \odot \Delta \odot \Theta, \ \mathbf{0}, \ \mathbf{1}\right\}, \tag{3}$$

where $\mathbf{N}_u \sim \mathcal{U}(-1, 1)$, $\mathbf{N}_g \sim \mathcal{N}(0, 1)$ and $\mathbf{N}_t \sim \mathcal{L}(0, 1)$. $\mathcal{L}$ denotes the Laplace distribution. $\alpha, \beta$ and $\gamma$ is the coefficient of $\mathcal{U}, \mathcal{N}$, and $\mathcal{L}$ distributions, respectively. Note that $\alpha + \beta + \gamma = 1$. $\Theta$ denotes the proposed dynamic varying noise ratio based on the detection area of the bounding box, and $\Theta = \tau \cdot (1 + \log(1 + \mathbf{A}))$. $\mathbf{A}$ denotes the area of the bounding box and $\mathbf{a} = \mathbf{w} \cdot \mathbf{h}$. Besides, $\mathbf{C}$ and $\Delta$ are defined as the same as the Equation 2. Our detailed algorithm is shown in Algorithm 1, and the mechanism proof of the Multi-Distributed Noise Mixing strategy is shown in **Appendix B**.

---

**Algorithm 1** **M**ulti-**D**istributed **N**oise **M**ixing Strategy

---

**Require:** Bounding box parameters $\mathbf{B}_e$, Noise scale $\tau$; Uniform noise weight $\alpha$, Gaussian noise weight $\beta$, Laplace noise weight $\gamma$

**Ensure:** Augmented bounding box parameters $\mathbf{B}_f$.

1: ▷ $\mathbf{D}$ is a tensor of the same shape as $\mathbf{B}_e$
2: Initialize $\mathbf{D} \leftarrow \mathbf{0}$
3: ▷ Compute the area of each bounding box
4: $\mathbf{A} \leftarrow \mathbf{B}_e[:,2] \cdot \mathbf{B}_e[:,3]$.
5: ▷ Compute dynamic noise scale for each box
6: $\mathbf{\Theta} \leftarrow \tau \cdot (1 + \log(1 + \mathbf{A}))$
7: ▷Define perturbation directions for center and size
8: $\mathbf{\Delta}[:,:2] \leftarrow \mathbf{B}_e[:,2:]/2$      # Perturb center
9: $\mathbf{\Delta}[:,2:] \leftarrow \mathbf{B}_e[:,2:]$        # Perturb width and height
10: ▷ Generate noise from multiple distributions
11: $\mathbf{N}_u \sim \mathcal{U}(-1,1)$, $\mathbf{N}_g \sim \mathcal{N}(0,1)$ and $\mathbf{N}_t \sim \mathcal{L}(0,1)$
12: ▷ Compute the weighted combination of noise
13: $\mathbf{N}_m \leftarrow \alpha \cdot \mathbf{N}_u + \beta \cdot \mathbf{N}_g + \gamma \cdot \mathbf{N}_l$
14: ▷ Add scaled noise to bounding box parameters
15: $\mathbf{B}_f \leftarrow \mathbf{B}_e + (\mathbf{N}_m \odot \mathbf{\Delta} \odot \mathbf{\Theta})$
16: ▷ Clamp augmented bounding boxes to the valid range
17: $\mathbf{B}_f \leftarrow \text{Clamp}(\mathbf{B}_f, \mathbf{0}, \mathbf{1})$
18: **return** $\mathbf{B}_f$

---

## 4.4 DYNAMIC MEMORY-GATED MODULE

To address the challenges of insufficient long-term dependency modeling and poor dynamic scene adaptation in our OVID task, we propose a Dynamic Memory-Gated Module. Given a input feature $\mathbf{X} \in \mathbb{R}^{\mathcal{B} \times \mathcal{C} \times \mathcal{H} \times \mathcal{W}}$, we first use global average pooling (GAP) to extract a global context query vector ($\mathbf{Q} \in \mathbb{R}^{\mathcal{B} \times \mathcal{C}}$), express it as $\mathbf{Q} = \text{GAP}(\mathbf{X})$, where $\mathbf{B}$, $\mathbf{C}$, $\mathbf{H}$ and $\mathbf{W}$ denotes the batch_size, channel, height and width. Then, we introduce a dynamic memory retrieval module and express it as

$$\mathbf{O}_m = \text{softmax}\left(\frac{\mathbf{Q}\mathbf{M}_K^{\mathbf{T}}}{\sqrt{d}}\right)\mathbf{M}_V, \tag{4}$$

where $\mathbf{Q} \in \mathbb{R}^{\mathcal{B} \times \mathcal{C}}$ denotes query vector. $\mathbf{M}_K \in \mathbb{R}^{\mathcal{B} \times \mathcal{C}}$ denotes the memory key, and $\mathbf{M}$ is the number of memory units. $\mathbf{M}_V \in \mathbb{R}^{\mathcal{M} \times \mathcal{C}}$ denotes the memory value, and $\mathbf{M}_V$ is used to store the feature information corresponding to the key. $\mathbf{O}_m \in \mathbb{R}^{\mathcal{B} \times \mathcal{C}}$ denotes the memory output by retrieving. Finally, retrieved memory output ($\mathbf{O}_m$) and input features ($\mathbf{X}$) are fused by concatenation and $1 \times 1$ Conv. Therefore, we can express this principle as

$$\mathbf{X}_f = \text{Conv1x1}(\text{Concat}(\mathbf{X} \odot \mathbf{W}, \mathbf{O}_m)), \tag{5}$$

where $\mathbf{X}_f$ denotes the fusion feature. $\mathbf{W}$ denotes the generate dynamic weights and $\mathbf{W} = \sigma(\mathbf{W}_2\text{ReLU}(\mathbf{W}_1\mathbf{Q}))$. $\mathbf{W}_1, \mathbf{W}_2$ denotes the weight of fully connected networks. $\mathbf{W}_1 \in \mathbb{R}^{\mathcal{C} \times d}$, $\mathbf{W}_2 \in \mathbb{R}^{d \times \mathcal{C}}$. $\sigma$ denotes the sigmoid function.

## 5 EXPERIMENTS AND ANALYSES

### 5.1 EXPERIMENTAL SETTINGS

**Implementation Details.** We conduct all experiments on a computer with 8 NVIDIA GeForce RTX 2080Ti GPUs. Unless specified, the Max_Iter, Batch_size_total, CHECKPOINT_PERIOD, EVAL_PERIOD of all experiments are set to 15000, 8, 15000, and 15000, respectively. The image encoder and text encoder adopt tiny-swin-transformer (Liu et al., 2021) and Clip (Radford et al., 2021), respectively. More hyperparameter details can be found in **Appendix C**.

**Datasets.** Our experiments are conducted in some publiy datasets, *e.g.*, COCO (Lin et al., 2014), Cityscape (Cordts et al., 2016a), Foggy-Cityscape (Sakaridis et al., 2018) and Cityintrusion-OpenV. In addition, to provide more visualization results, we also test and report visualization demo results

on other datasets, *e.g.*, the ShanghaiTech Campus dataset (Luo et al., 2017), and the UA-DETRAC (Wen et al., 2020). Note that in our experiment, we adopt two manners, *i.e.*, zero-shot and task-specific transfer, to evaluate the performance of the model.

**Metrics.** In order to report the quantitative results of our experiments more comprehensively, inspired by some previous promising work (Han et al., 2024c;b), the mIOU(%) and mAP(%) are utilized to evaluate the sub-task performance of segmentation and object detection. For the intrusion detection performance, we also use three different intrusion detection metrics: AccY(%), AccN(%), and Acc(%) to quantify. Besides, some additional metrics, *e.g.*, panoptic segmentation metrics, PQ(%), SQ(%), RQ(%), and every AP(%), AP@.5(%) of intrusion categories, are reported to evaluate the zero-shot performance of the model.

**Comparison Models.** We compare with the OpenSeeD (Zhang et al., 2023) because of its promising multi-task capability and performance in open-vocabulary tasks. The multi-task feature is consistent with our task. Besides, we also compare the latest intrusion works, *e.g.*, PIDNet (Sun et al., 2020), Cross-PIDNet (Shi et al., 2022), MMID-bench (Han et al., 2024c), MF-ID (Han et al., 2024b).

## 5.2 MAIN RESULTS

**Compared with promising open-vocabulary works.** We first compare the multiple performances with the promising OpenSeeD model and report three zero-shot detection performances, *i.e.*, PQ(%), SQ(%), RQ(%), and three task-specific transfer intrusion performances, *i.e.*, AccY(%), AccN(%), Acc(%), as shown in Table 2. We can find that in different tasks, for the panoptic segmentation performance (PQ), compared with OpenSeeD, our methods can improve it by 2.19% and 1.12%, respectively. Besides, for intrusion detection performance (Acc), our model can surpass it by 3.43% and 3.45%, respectively, which verifies the effectiveness of the proposed model and strategies.

Table 2: The zero-shot and task-specific transfer comparison results between promising multi-task open-vocabulary work and OVIDNet in different datasets. More results are shown in **Appendix D.**

| - | Zero-shot Detection (Panoptic segmentation) | | | Task-specific Transfer (Intrusion detection) | | | |
|---|---|---|---|---|---|---|---|
| Model | Test data 1 | RQ(%) | SQ(%) | PQ(%) | Test data 2 | AccY(%) | AccN(%) | Acc(%) |
| OpenSeeD | Cityscape | 18.22 | 43.68 | 14.03 | Ours (Normal) | 18.72 | 36.19 | 29.36 |
| | Foggy-Cityscape | 18.07 | 36.71 | 14.28 | Ours (Foggy) | 22.04 | 25.88 | 24.38 |
| OVIDNet (Ours) | Cityscape | 20.36 | 36.17 | **16.22** | Ours (Normal) | 24.43 | 38.16 | **32.79** |
| | Foggy-Cityscape | 19.05 | 33.71 | **15.40** | Ours (Foggy) | 27.72 | 27.90 | **27.83** |

**Compared with promising intrusion detection works.** We also compare some intrusion detection works, as shown in Table 3. Note that the detailed results of our model can be seen in Tabel 14 of **Appendix E.1**. We can see that, compared with previous intrusion works, our model not only has an open structure but also detects more intrusion categories. More importantly, our model has strong generalization capability and achieves zero-shot detection, which is not only pre-trained/pre-undefined categories. In addition, we can observe that as the task difficulty increases, *i.e.*, common intrusion detection task (PIDNet, Cross-PIDNet, MF-ID)→domain adaptation intrusion detection task (MMID-bench)→Open-vocabulary intrusion detection task (OVID), the performance of each category continuously decreases. The main reason is that, as the difficulty of different intrusion detection tasks increases, the requirements of different intrusion detection frameworks are also raised in the open world, especially their generalization and zero-shot capabilities.

Table 3: The comparison between our work and promising intrusion detection works. 'close' and 'open' denote the different detection structures. 'ZSD' denotes Zero-shot detection. ✓ and ✗ denote the intrusion category as assessable or not assessable, respectively. † denotes that the backbone is BNet.

| Method | Venue | Structure | ZSD | P(%) | R(%) | M(%) | Bc(%) | Tk(%) | Bu(%) | Tn(%) | C(%) |
|---|---|---|---|---|---|---|---|---|---|---|---|
| PIDNet (Sun et al., 2020) | ACM MM'20 | close | ✗ | 67.1 | ✗ | ✗ | ✗ | ✗ | ✗ | ✗ | ✗ |
| | | | ✗ | 63.3† | ✗ | ✗ | ✗ | ✗ | ✗ | ✗ | ✗ |
| Cross-PIDNet (Shi et al., 2022) | T-IV'21 | close | ✗ | 74.7 | ✗ | ✗ | ✗ | ✗ | ✗ | ✗ | ✗ |
| | | | ✗ | 72.1† | ✗ | ✗ | ✗ | ✗ | ✗ | ✗ | ✗ |
| MF-ID (Han et al., 2024b) | T-ASE'24 | close | ✗ | 45.8 | 39.8 | 34.5 | 38.2 | ✗ | ✗ | ✗ | ✗ |
| MMID-bench (Han et al., 2024c) | T-IV'24 | close | ✗ | 37.4 | 34.6 | 20.7 | 33.1 | ✗ | ✗ | ✗ | ✗ |
| OVIDNet (Ours) | - | open | ✓ | ✓ | ✓ | ✓ | ✓ | ✓ | ✓ | ✓ | ✓ |

**Zero-shot and Task-specific transfer evaluation results on proposed strategies.** We then test the zero-shot/task-specific transfer performance of the proposed strategies. Specifically, we train our model on the COCO dataset and validate it on the Cityscape datasets to ob-

Table 4: Zero-shot and Task-specific transfer quantitative results of the proposed different strategies.

| B | DMG | MDNM | PQ(%) | mIOU(%) | mAP@.5(%) | AccY(%) | AccN(%) | Acc(%) |
|---|-----|------|-------|---------|-----------|---------|---------|--------|
| ✓ | ✗ | ✗ | 14.03 | 28.34 | 27.58 | 18.72 | 36.19 | 29.36 |
| ✓ | ✓ | ✗ | 15.80 | 28.78 | 29.16 | 20.06 | 37.56 | 30.72 |
| ✓ | ✗ | ✓ | 15.33 | 29.40 | 28.56 | 21.01 | 38.64 | 31.75 |
| ✓ | ✓ | ✓ | 16.22 | 29.37 | 28.98 | 24.43 | 38.16 | **32.79** |

tain the segmentation and detection performance in a zero-shot manner. Besides, we also test the intrusion detection performance on the proposed Cityintrusion-OpenV datasets by a task-specific transfer manner, as shown in Table 4. **B** denotes the baseline. We can observe that as different strategies are added, multiple performances are improved, not only intrusion detection but also zero-shot performance, *e.g.*, PQ(%) and mIOU(%). Compared with the baseline, the intrusion performance (Acc) can surpass it by 3.43%. In addition, the zero-shot performance can surpass it by 2.19% (PQ) and 1.03% (mIOU), respectively. More detailed results can be found in **Appendix E.1**.

**Generalization Verification in cross-domain tasks.** We further test the performance of our OVID-Net framework and strategies in cross-adverse weather tasks, *e.g.*, Normal→Foggy, to verify generalization capabilities. Note that all performance results are given by the pre-trained (in Normal weather) and inference (in adverse weather) manners, as shown in Figure 4. We can find that our OVIDNet is effective even in adverse weather and exhibits promising intrusion performance. Under three different foggy coefficient setting, *i.e.*, $\alpha = 0.005$, $\alpha = 0.01$, $\alpha = 0.02$, our OVIDNet can surpass the baseline model by 2.96%, 3.22%, and 3.45%, respectively. Besides, our strategies can also improve the zero-shot performance under cross-domain tasks, *e.g.*, compared with the baseline of three different foggy coefficient settings, the PQ(%) in the Normal→Foggy tasks can surpass them by 1.29%, 1.21%, and 1.12%, respectively. More details results can be found in **Appendix E.2**.

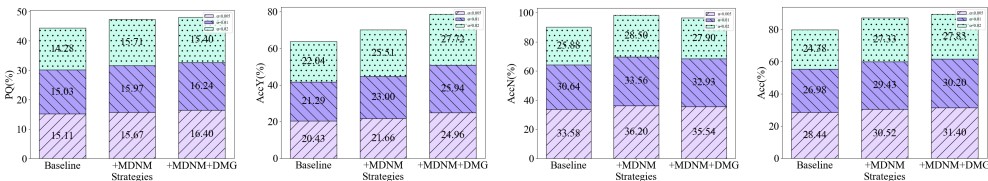

Figure 4: Generalization Verification in cross-domain tasks.

**Visualization Comparisons.** We also present some visualization comparison results to verify the zero-shot performance and effectiveness of the proposed framework and methods, as shown in Figure 5. We can find that our framework can present promising visualization detection results, not only detecting intrusion behaviors correctly but also giving correct Intrusion ('Y')/No-intrusion ('N') labels, which proves the effectiveness of our framework and approach. Note that our OVIDNet can improve the zero-shot segmentation performance of AoI; in this case, the AoI is the road. More visualization comparison results are presented in **Appendix E.3**.

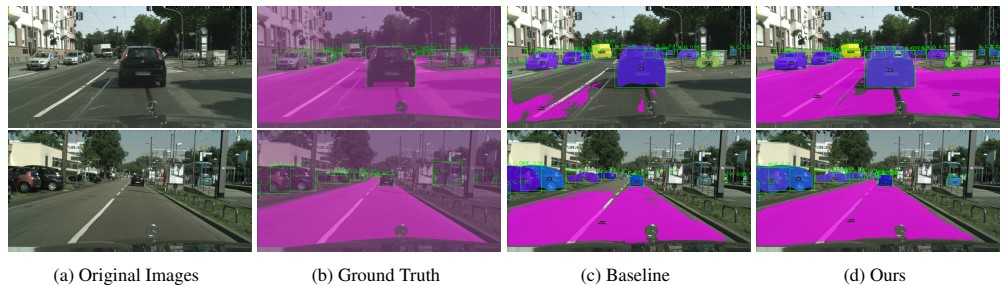

| (a) Original Images | (b) Ground Truth | (c) Baseline | (d) Ours |

Figure 5: The visualization comparison results.

## 5.3 ABLATION EXPERIMENTS

**Multi-Distributed Noise Mixing Strategy.** We analyze the proposed multi-distributed noise mixing strategy and conduct extra ablation experiments to verify its effectiveness. The detailed results

are shown in Table 5. We can find that when the $\alpha$=**0.5**, $\beta$=**0.1**, $\gamma$=**0.4**, the intrusion detection performance can reach the best, with a 31.75% intrusion accuracy. The main reason is that, in task-specific transfer, the model focuses more on texture features and spatial perturbations. Besides, the transfer task is performed during normal weather. Thus, the need for weather changes and light perturbations is low. In this paper, we set the $\alpha$, $\beta$, $\gamma$ to **0.5**, **0.1** and **0.4**, respectively.

**Dynamic Memory-Gated Module.** We also explore the effect of different memory unit sizes on intrusion detection performance, as shown in Table 5. $IOU^r$ denotes the zero-shot segmentation results of the road. We can see that the best intrusion performance can be reached when **M=40**. The main reason is that the larger memory can help capture richer history and global features, especially in open-world intrusion detection. However, larger memory units also introduce more irrelevant information, making it difficult to focus on key memory features. Conversely, fewer memory units can help the model focus on features relevant to intrusion detection, but if the memory units are too low, it will lead to a loss of diversity and complexity required for the intrusion task, affecting the understanding of complex intrusion scenarios. In this paper, we set the memory units to **40**.

Table 5: The ablation experiments of the proposed strategies. **B** denotes the baseline.

| Ablation 1: Multi-Distributed Noise Mixing Strategy | | | | | | | |
|---|---|---|---|---|---|---|---|
| **B** | #$\mathcal{U}(\alpha)$ | #$\mathcal{N}(\beta)$ | #$\mathcal{L}(\gamma)$ | PQ | $IOU^r$(%) | mAP(%) | Acc(%) |
| ✓ | ✓(1) | ✗ | ✗ | 14.03 | 74.1 | 27.6 | 29.36 |
| ✓ | ✓(0.5) | ✓(0.4) | ✓(0.1) | 14.37 | 70.1 | 32.1 | 30.28 |
| ✓ | ✓(0.5) | ✓(0.2) | ✓(0.3) | 14.78 | 68.8 | 25.8 | 30.48 |
| ✓ | ✓(0.5) | ✓(0.3) | ✓(0.2) | 14.53 | 78.5 | 25.9 | 31.03 |
| ✓ | ✓(0.5) | ✓(0.1) | ✓(0.4) | 15.33 | 74.6 | 28.6 | **31.75** |
| Ablation 2: Dynamic Memory-Gated Module | | | | | | | |
| **B** | Memeroy units | | | PQ | $IOU^r$(%) | mAP(%) | Acc(%) |
| ✓ | ✗ | | | 14.03 | 74.1 | 27.6 | 29.36 |
| ✓ | ✓(M=30) | | | 14.84 | 76.0 | 27.5 | 30.31 |
| ✓ | ✓(M=40) | | | 15.80 | 76.5 | 29.5 | **30.72** |
| ✓ | ✓(M=50) | | | 15.27 | 80.1 | 27.9 | 30.34 |

## 5.4 MORE INSIGHTFUL AND INTERESTING EXPERIMENTS

**Experiment 1: Why is the performance result of category 'Rider' is '0.0'?**

In some table, we find that the performance of category 'Rider' is '**0.0**', *e.g.*, Table 14 and Table 15. To answer this question, 1) we first investigate some of the latest open-vocabulary works (Bianchi et al., 2024; Ma et al., 2025). Some works denote that the understanding of fine-grained properties of objects and their parts is important. From this view, we conduct some experiments and provide the visualization comparison, as shown in Figure 6. We can find that our model recognizes the category 'Rider' as the category 'Person'. The main reason is that these two categories have similar features. 2) Besides, in the training dataset, the number of category 'Person' is much larger than the category 'Rider,' which leads to category imbalance. Therefore, these two factors will make it difficult to recognize the fine-grained category 'Rider'. To compensate for this gap, we design a simple yet effective reasoning enhancement strategy, *i.e.*, Geometric Constraint Reclassification Strategy (GCRS). The detailed princlples of GCRS is shown in **Appendix F**. Then, we conduct several experiments to verify the effectiveness of the GCRS strategy, as shown in Table 6. We can find that when applying the proposed strategy, the performance of the rider category improved from 0 to 23.56. Additionally, intrusion detection performance has improved, *i.e.*, 32.79→33.55.

Table 6: The performance of GCRS.

| Performance | Rider_Y | Rider_N | Rider | Acc |
|---|---|---|---|---|
| OVIDNet (Ours) | 0 | 0 | 0 | 32.79 |
| + GCRS | 11.95 | 31.52 | 23.56 | **33.55** |

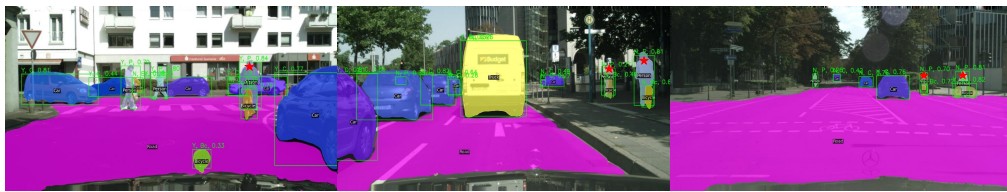

Figure 6: Some cases of recognizing 'Rider (R)' as 'Person (P)'. ★ denotes the case locations.

**Experiment 2: Real-scenario application exploration.** To verify the high generality and universal applicability of our framework, we also provide some visualization results under different static scenarios, *e.g.*, intelligent monitoring, and security. Since static scenarios lack relevant intrusion detection datasets and labels, the specific quantitative evaluation results cannot be measured and given. However, inspired by some super-resolution works (Gandikota & Chandramouli, 2024; Korkmaz et al., 2024), we can report some demo visualization results. Here, we directly use our framework

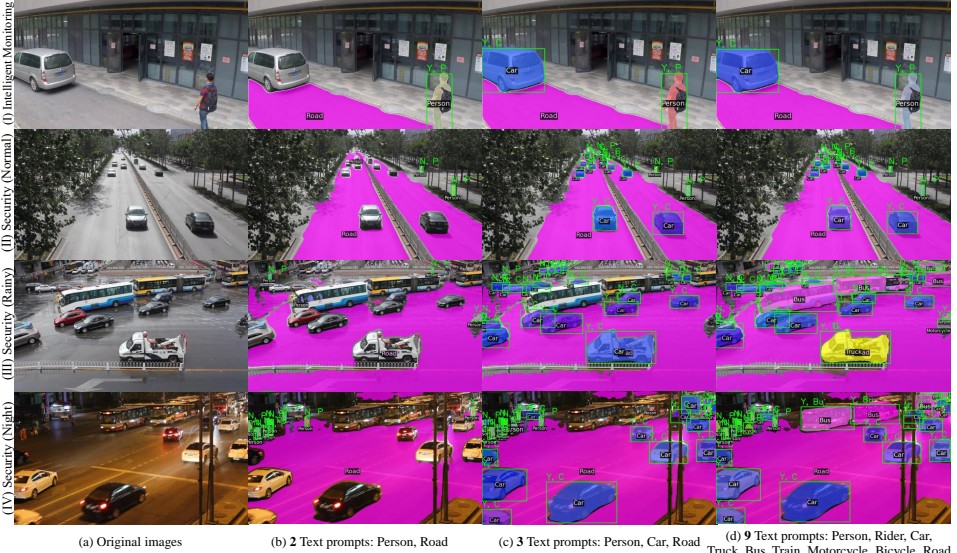

(a) Original images      (b) **2** Text prompts: Person, Road      (c) **3** Text prompts: Person, Car, Road      (d) **9** Text prompts: Person, Rider, Car, Truck, Bus, Train, Motorcycle, Bicycle, Road

Figure 7: The visualization demo results in real scenarios. We directly utilize our framework to infer public static scenario datasets without any retraining process. We give three different text prompts customizable results, *i.e.*, 2 text prompts, 3 text prompts, and 9 text prompts, respectively.

to infer public static scene datasets without any retraining process, *e.g.*, the ShanghaiTech Campus dataset (Luo et al., 2017), and the UA-DETRAC (Wen et al., 2020). Note that for different scenarios and intrusion categories, the number of text prompts can be customized. In our paper, we report multiple visualization results with different customizable text prompts . Besides, we use different domains, *e.g.*, Normal, Rainy, and Night, to evaluate the model's generalization performance, as shown in Figure 7. We can observe that our framework can detect and judge intrusion behavior, demonstrating the practicality and effectiveness of the proposed framework.

## 6 DISCUSSIONS

To consider more distribution shift types and enhance the diversity of intrusion scenes in open-world deployment, we created a new intrusion detection dataset for the OVID task, namely Cityintrusion-OpenV-BDD. The new dataset is built based on the BDD-100K datasets (Yu et al., 2020). The detailed information can be found in **Appendix G**. Our new datasets contain rich intrusion scene types.

Table 7: The detailed experimental results on Cityintrusion-OpenV-BDD dataset.

| Baseline | DMG | MDNM | AccY(%) | AccN(%) | Acc(%) | Gain |
|----------|-----|------|---------|---------|--------|------|
| ✓ | ✗ | ✗ | 20.99 | 17.22 | 18.69 | - |
| ✓ | ✓ | ✗ | 20.16 | 19.27 | 19.62 | +0.93 |
| ✓ | ✗ | ✓ | 20.26 | 20.38 | 20.33 | +1.64 |
| ✓ | ✓ | ✓ | 25.79 | 21.66 | **23.27** | **+4.58** |

We evaluate the performance of our model on the datasets, as shown in the Table 7. We can find that, in different domain shifts, our strategies still present promising performance improvements. Compared with the baseline, our model can surpass it by **4.58%**, which verifies the strong robustness of our model and the effectiveness of the proposed strategies.

## 7 CONCLUSION

In this paper, we propose a new and vital intrusion detection task, Open-Vocabulary Intrusion Detection (OVID). This is the first multi-modal attempt in the vision-based intrusion detection task. A new benchmark, including a relative dataset, an efficient multi-modal framework, and some strong baselines, is given for the specific task. Besides, two effective strategies are proposed to improve the generalization and enhance the performance of the intrusion detection task in open scenarios, *i.e.*, the Multi-Distributed Noise Mixing and the Dynamic Memory-Gated module. Finally, rich experiments and comparisons are done to demonstrate the effectiveness of the proposed framework and strategies. In the future, we will further explore more useful methods to improve performance.

ACKNOWLEDGMENTS

This work was supported by a locally commissioned task from the Shanghai Municipal Government and the Shanghai Artificial Intelligence Laboratory (AILab). We also express our gratitude to our collaborators (Fujun Han and Jingqi Ye) for their valuable contributions during their internships at AILab, both in-person and remotely. We sincerely thank all authors for their contributions.

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

**APPENDIX OVERVIEW**

**Table of contents:**

# A CITYINTRUSION-OPENV DATASET

In this subsection, we present additional information and details for the proposed Cityintrusion-OpenV dataset, including data statistics, visualization results, intrusion Dataset Comparisons, and the correspondence between full name, text prompt, abbreviation, and framework design motivation.

## A.1 METHOD AND STATISTICS FOR PROPOSED DATASET

**Automatic Generation Method**. Inspired by promising intrusion detection works (Sun et al., 2020; Shi et al., 2022; Han et al., 2024b), our Cityintrusion-OpenV dataset is built based on the Cityscape dataset (Cordts et al., 2016a). The main reason is that the Cityscape datasets have segmentation and detection labels for the same original image, which provides a prerequisite for our multiple intrusion detection tasks. Additionally, following the relevant works (Han et al., 2024c), we also design an automatic labeling program to generate Intrusion ('Y') and No-intrusion ('N') labels. Note that the final intrusion detection overlapping pixel points are also set to **20** (Sun et al., 2020). The specific processes are shown as follows.

· **Step 1:** We first clean the original Cityscape (Cordts et al., 2016b)/Foggy-Cityscape (Sakaridis et al., 2018). After cleaning, we conduct frame alignment for these datasets. Note that a small number of objects that we don't care about or are incorrectly labeled will be removed in this process.

· **Step 2:** Based on the results in step 1, we can read the bounding box coordinates of the interested intrusion objects from the Cityscape/Foggy-Cityscape datasets. Additionally, we also read the area-of-interest (AoI) in the Cityscapes segmentation dataset (Cordts et al., 2016b).

· **Step 3:** For the obtained area-of-interest (AoI) in step 2, we binarize them with **0** and **1**.

· **Step 4:** After step 3, the bounding box coordinates from step 2 are projected into the binarized area-of-interest (AoI).

· **Step 5:** We calculate the overlapping pixel values between AoI and bounding box in step 4.

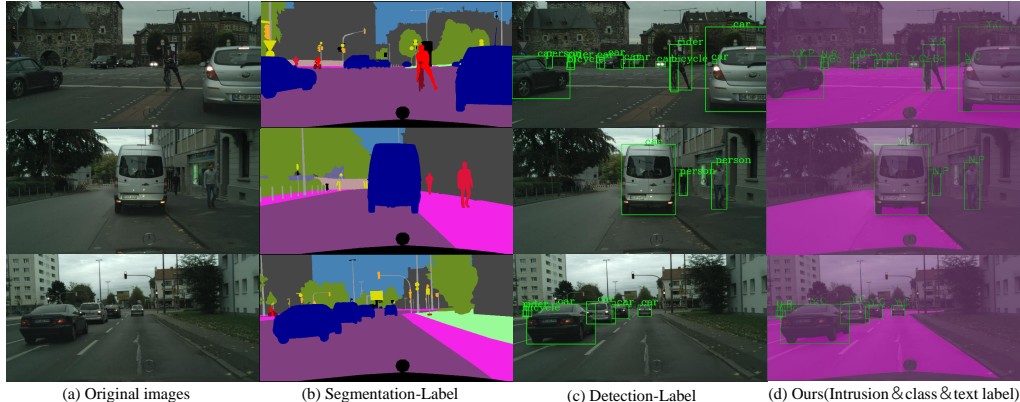

(a) Original images   (b) Segmentation-Label   (c) Detection-Label   (d) Ours(Intrusion&class&text label)

Figure 8: More visualization results of our Cityintrusion-OpenV. Unlike previous intrusion detection datasets (Sun et al., 2020; Han et al., 2024b), our datasets encompass all common/possible intrusion categories in Cityscape datasets, providing richer and varied labels that meet the requirements of the proposed OVID task.

· **Step 6:** To get the final intrusion/no-intrusion labels: 'N/Y, Class', we compare overlapping pixel values in step 5 with a setting threshold, where 'N' denotes Non-Intrusion, 'Y' denotes Intrusion, and 'Class' denotes names of intrusion objects. Note that, following previous work (Sun et al., 2020), the threshold is set to 20.

· **Step 7:** To obtain and present our final intrusion detection dataset better, we blended the segmented images containing the intrusion labels in step 6 with the original images in step 1.

· **Step 8:** Finally, to ensure the quality and accuracy of the proposed datasets, a team of three students are organized to manually inspect and verify the annotations.

**Statistical Analysis**. Then, we conduct a detailed statistical analysis, as shown in Table 8. We provide details of the number of intrusion and non-intrusion cases for each category in the training and validation sets of the dataset, along with the total average. The total average of the whole dataset can reach **18.03**, surpassing previous promising intrusion detection datasets greatly. Rich labels can meet the requirements and provide a data foundation for the proposed OVID task.

Table 8: The detailed statistics of proposed datasets. T and V denote the training and validation datasets, respectively. ‡ denotes the total average in the whole dataset.

| Categories | Person | Rider | Car | Truck | Bus | Train | Motorcycle | Bicycle |
|---|---|---|---|---|---|---|---|---|
| | T\|V | T\|V | T\|V | T\|V | T\|V | T\|V | T\|V | T\|V |
| Intrusion cases ('Y') | 3567\|716 | 698\|226 | 14545\|2493 | 246\|52 | 219\|67 | 88\|9 | 270\|55 | 1138\|361 |
| Non-Intrusion cases ('N') | 14427\|2703 | 1109\|330 | 12610\|2174 | 243\|41 | 166\|31 | 83\|14 | 469\|94 | 2591\|814 |
| Total cases | 17994\|3419 | 1807\|556 | 27155\|4667 | 489\|93 | 385\|98 | 171\|23 | 739\|149 | 3729\|1175 |
| Total average‡ (Per very image) | **18.03** | | | | | | | |

## A.2 MORE VISUALIZATION RESULTS

In order to better present our proposed dataset, we also provide more visualization results, as shown in Figure 8. Different from the previous single category (President) (Sun et al., 2020; Shi et al., 2022) and four categories (President, Motorcycle, Rider, Bicycle) (Han et al., 2024b;c), we can find that our Cityintrusion-OpenV dataset contains multiple different intrusion categories, not only single or four categories. All possible intruder categories can be considered in our dataset, *e.g.*, Person, Rider, Car, Truck, Bus, Train, Motorcycle, Bicycle. Our new dataset can provide the prerequisite for the OVID task. Here, because of the labels, we utilize abbreviations instead of labels in order to easily show our results, *e.g.*, 'N, P' denotes the 'No-Intrusion, Person', Text prompt: Person. 'Y, C' denotes the 'Intrusion, Car', Text prompt: Car. 'Y, Bu' denotes the 'Intrusion, Bus', Text

Table 9: The correspondence between the full name, text prompt, and abbreviation. *Italic* denotes *thing classes* (AoI). All categories are customizable in different scenarios.

| # No. | # Full name | # Text prompt | # Abbreviation |
|-------|-------------|---------------|----------------|
| # 1 | 'Person' | 'Person' | 'P' |
| # 2 | 'Rider' | 'Rider' | 'R' |
| # 3 | 'Car' | 'Car' | 'C' |
| # 4 | 'Truck' | 'Truck' | 'Tk' |
| # 5 | 'Bus' | 'Bus' | 'Bu' |
| # 6 | 'Train' | 'Train' | 'Tn' |
| # 7 | 'Motorcycle' | 'Motorcycle' | 'M' |
| # 8 | 'Bicycle' | 'Bicycle' | 'Bc' |
| # 9 | *'Road'* | *'Road'* | *'Ro'* |
| *# 10* | ⋮ | ⋮ | ⋮ |

prompt: Bus. 'Y, M' denotes the 'Intrusion, Motorcycle', Text prompt: Motorcycle. The detailed correspondence between full name, text prompt, and abbreviation and can be found in Table 9.

## A.3 MORE INTRUSION DATASET COMPARISONS

In this subsection, we further compare our proposed datasets with other promising intrusion detection datasets and provide more comparison results to verify the superiority of our dataset, as shown in Figure 9. Compared to previous promising intrusion detection datasets (Sun et al., 2020; Han et al., 2024b;c), our dataset exhibits much superior and richer labels. Besides, the proposed datasets contain **8** intrusion categories, surpassing the previous works **1** or **4** categories. More importantly, our Cityintrusion-OpenV dataset contains text labels, which compensate for the lack of relevant datasets and meet the needs of the proposed OVID task.

## A.4 THE CORRESPONDENCE BETWEEN FULL NAME, TEXT PROMPT, AND ABBREVIATION

To better help understand the different intrusion categories and the abbreviations in our paper, we provide the detailed correspondence between the full name, the text prompt, and the abbreviation. The detailed correspondence is shown in the Table 9, *e.g.*, 'Person' (# Full name) → 'Person' (# Text prompt) → 'P' (# Abbreviation), 'Rider' (# Full name) → 'Rider' (# Text prompt) → 'R' (# Abbreviation), 'Car' (# Full name) → 'Car' (# Text prompt) → 'C' (# Abbreviation), 'Truck' (# Full name) → 'Truck' (# Text prompt) → 'Tk' (# Abbreviation), 'Bus' (# Full name) → 'Bus' (# Text prompt) → 'Bu' (# Abbreviation).

## A.5 FRAMEWORK DESIGN MOTIVATION

In this subsection, We first explore two basic yet important questions as motivations for our approach. (1) **Why** do we conduct open vocabulary intrusion detection research? Our goal is to break through the dependencies and limitations of pre-defined categories. Truly enable intrusion detection in the open world. (2) **How** to achieve the specific OVID task? A simple idea is that we can leverage a collaborative model with Open-vocabulary segmentation (OVS), *e.g.*, SAM (Kirillov et al., 2023), FastSAM (Zhao et al., 2023), EfficientSAM (Xiong et al., 2024), and Open-vocabulary detection (OVD), *e.g.*, DetClip (Yao et al., 2023), Grounding DINO (Liu et al., 2024), YOLO-world (Cheng et al., 2024) to train/infer and get final Intrusion/No-intrusion labels. As shown in Table 10, we list and compare some feasible schemes. Unfortunately, although the model of 'OVD+OVS' is a feasible solution, it is not suitable for intrusion detection. The main reason is that the training cost of the End-to-End strategy combined with two **LLVMs** (Large Language Vision Models) is very expensive. To alleviate this problem, we design a new efficient framework for the proposed OVID task, namely OVIDNet. Our framework is established based on OpenSeeD (Zhang et al., 2023). Finally, the OVIDNet is leveraged to collaborate to give the bounding box and mask image for the OVID

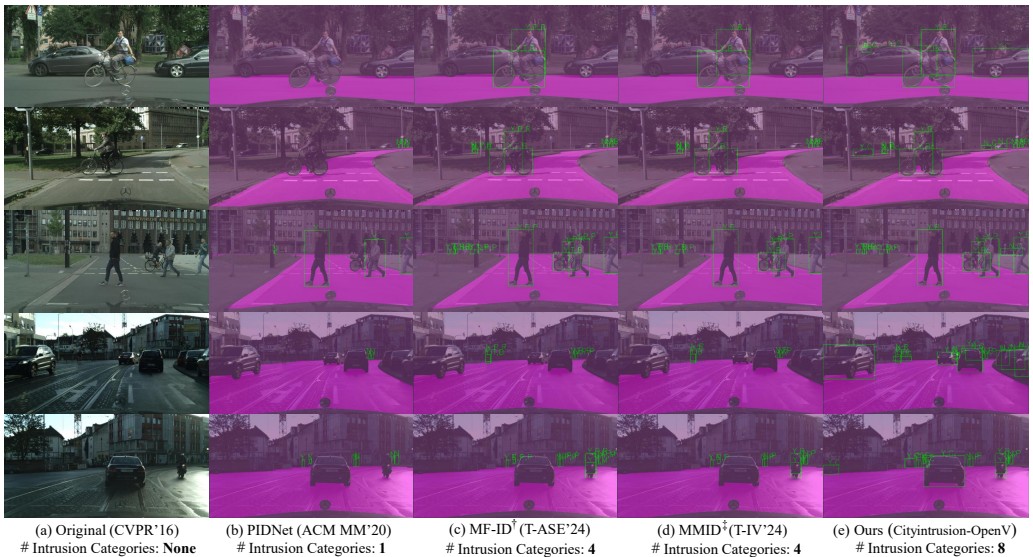

| (a) Original (CVPR'16) | (b) PIDNet (ACM MM'20) | (c) MF-ID† (T-ASE'24) | (d) MMID‡ (T-IV'24) | (e) Ours (Cityintrusion-OpenV) |
| # Intrusion Categories: **None** | # Intrusion Categories: **1** | # Intrusion Categories: **4** | # Intrusion Categories: **4** | # Intrusion Categories: **8** |

Figure 9: The compression between our datasets with other promising intrusion detection datasets. Unlike previous intrusion datasets (Sun et al., 2020; Han et al., 2024b), our datasets encompass a broader range of potential intruders. Our datasets can be used to train/evaluate the performance of the OVID task and validate the effectiveness of the proposed strategies.

task, and the final intrusion labels ('N/Y') are given by the intrusion post-processing judgments. The overall framework and pipeline of OVIDNet are illustrated below.

Table 10: The comparison of some feasible schemes for the proposed OVID task. OVD and OVS denote the Open-vocabulary detection and segmentation models. Pre-trained TSM denotes the pre-trained traditional segmentation models, *e.g.*, DeepLabv3+ (Chen et al., 2018), PspNet (Zhao et al., 2017). Retrain denotes whether the model needs to be retrained under different scenarios. We can find that our scheme is End-to-End and has a low training cost.

| Scheme | OVD Train | OVD Infer | OVS Train | OVS Infer | Pre-trained TSM | End-to-End | Open-Vocabulary? | Retrain | Training Cost |
|---|---|---|---|---|---|---|---|---|---|
| S1 | ✓ | ✓ | ✓ | ✓ | ✗ | ✗ | ✓ | ✗ | Very Large |
| S2 | ✓ | ✓ | ✗ | ✗ | ✓ | ✓ | ✗ | ✓ | Large |
| S3 | ✓ | ✓ | ✗ | ✓ | ✗ | ✗ | ✓ | ✗ | Low |
| Ours | ✓ | ✓ | ✓ | ✓ | ✗ | ✓ | ✓ | ✗ | Low |

## B MECHANISM PROOF FOR MULTI-DISTRIBUTED NOISE MIXING STRATEGY

To clarify why the proposed noise mixture improves generalization toward unknown categories, we provide a theoretical proof based on Vicinal Risk Minimization (VRM) (Chapelle et al., 2000). In Equation 3, our perturbation can be written as

$$\mathbf{B}_f = \mathcal{C}\left\{\mathbf{B}_e + (\alpha \cdot \mathbf{N}_u + \beta \cdot \mathbf{N}_g + \gamma \cdot \mathbf{N}_t) \odot \Delta \odot \Theta, \mathbf{0}, \mathbf{1}\right\}. \tag{6}$$

To facilitate subsequent derivation, we set $Z = \alpha \cdot \mathbf{N}_u + \beta \cdot \mathbf{N}_g + \gamma \cdot \mathbf{N}_t$. And $\mathbf{N}_u, \mathbf{N}_g, \mathbf{N}_t$ denote noise sampled from Uniform, Gaussian, and Laplace distributions, respectively. Since $Z$ is a combination of three independent noise sources, its distribution can be expressed as

$$p_Z(z) = \alpha p_u(z) + \beta p_g(z) + \gamma p_t(z), \tag{7}$$

which is a mixed distribution containing (1) fine-grained bounded perturbations (U), (2) moderate Gaussian variations (Gaussian), (3) heavy-tailed structural deviations (Laplace). This directly yields a mixed vicinal distribution in the Vicinal Risk Minimization (VRM) framework. Following previous work (Chapelle et al., 2000), the vicinal risk can be written as

$$\hat{R}_{\text{VRM}}(f) = \mathbb{E}_{(x,y)\sim\mathcal{D}}\mathbb{E}_{z\sim p_Z}\ell(f(x+z), y). \tag{8}$$

Because $p_Z$ is a mixture, VRM can be expressed as

$$\hat{R}_{\text{M-VRM}}(f) = \alpha\hat{R}_u(f) + \beta\hat{R}_g(f) + \gamma\hat{R}_t(f), \tag{9}$$

where $\hat{R}_u(f) = \mathbb{E}_{z\sim p_u}\ell(f(x+z), y), \hat{R}_g(f) = \mathbb{E}_{z\sim p_g}\ell(f(x+z), y), \hat{R}_t(f) = \mathbb{E}_{z\sim p_t}\ell(f(x+z), y)$. Thus, instead of training against a single perturbation model, our model effectively optimizes risk under three complementary vicinal neighborhoods simultaneously.

Let the real open-world perturbation distribution be $q(z)$. If $q$ and $p_Z$ share support and $q \ll p_Z$, then the Radon-Nikodym ratio can be expressed as

$$K := \sup_z \frac{q(z)}{p_Z(z)}, \tag{10}$$

and the $K$ is finite, giving

$$q(z) \leq Kp_Z(z). \tag{11}$$

Because $p_Z(z) = \alpha p_u(z) + \beta p_g(z) + \gamma p_t(z)$ covers bounded signals (U), smooth variations (Gaussian), and heavy-tailed structural changes (Laplace). It has wider support and a smaller worst-case ratio $K$ than any single distribution. Therefore, the true risk is upper-bounded by

$$R_q(f) = \mathbb{E}_{z\sim q}\ell(f(x + z), y) \leq K\hat{R}_{\text{M-VRM}}(f). \tag{12}$$

A smaller constant $K$ implies a tighter bound. Hence, our model can better generalize to unknown object shapes, sizes, and distributions, and improve localization for novel or rare categories. Because their perturbations are more likely to fall inside the large support of the mixture $p_Z$.

## C    MORE EXPERIMENT SETTINGS

In this section, we will introduce more implementation details and settings in the experiments. We present more setting details for the experiment, as shown in Table 11. Due to the limitation of our GPUs, we have to set the Image_size to 800 and reduce the iterations to 15000, which inevitably makes some of our results lower than those in the original model (image_size: 1200×1200, max_iter: 368750) (Zhang et al., 2023). To ensure fairness and verify the correctness of our method, we also set CHECKPOINT_PERIOD and EVAL_PERIOD to 15000, respectively, for all experiments. Additionally, we retrain the baseline and verify the method's validity. Note that our OVIDNet framework is built based on the OpenSeeD (Zhang et al., 2023). The OpenSeeD is a simple but efficient framework for open-vocabulary segmentation and detection. Differently, we modify the original framework to meet the requirements of the OVID task. We first add an effective intrusion detection judgment module to obtain the capability for intrusion detection. Then, we propose two strategies for improving generalization and intrusion performance in the open world and verify the effectiveness on multiple dominant datasets and tasks. Therefore, to be fair, our setting mainly refers to Openseed and is adapted to our own tasks. Our OVIDNet framework consists of a text encoder (Clip (Radford et al., 2021)), image encoder (Tiny-swin-transformer (Liu et al., 2021)), decoder, and intrusion detection post-processing module. Note that the final overlapping pixel threshold is set to 20 (Sun et al., 2020).

## D    THE COMPARISONS WITH OTHER STRONG OPEN-VOCABULARY DETECTION SYSTEMS

To further validate the effectiveness of our model, we conduct a thorough investigation for other open-vocabulary detection models, *e.g.*, YOLO-World (Cheng et al., 2024) and Grounding DINO (Liu et al., 2024), and find that they primarily focus on the open-vocabulary detection domain. However, our task requires not only detection sub-tasks but also segmentation sub-tasks and intrusion detection sub-tasks. Therefore, open-vocabulary detection models alone cannot fulfill our intrusion detection requirements. Besides, we also find that some models support the open-vocabulary instance segmentation. However, these models also cannot meet the needs of the OVID task, as shown in Table 12. Nevertheless, to validate the effectiveness of our proposed model, we conducted some experiments on the YOLO-world and GroundingDINO models. Specifically, we follow previous intrusion detection work (Sun et al., 2020; Han et al., 2024b) and adopt a combined

Table 11: The detailed illustration of the experiment setting.

| # Name 1 | Setting Category 1 | Value | # Name 2 | Setting Category 2 | Value |
|---|---|---|---|---|---|
| TOKENIZER | | CLIP | WINDOW_SIZE | | 7 |
| CONTEXT_LENGTH | | 18 | PATCH_SIZE | | 4 |
| WIDTH | TEXT | 512 | EMBED_DIM | BACKBONE | 96 |
| HEADS | | 8 | DEPTHS | | [ 2, 2, 6, 2] |
| LAYERS | | 12 | NUM_HEADS | | [ 3, 6, 12, 24 ] |
| # Name 3 | Setting Category 3 | Value | # Name 4 | Setting Category 4 | Value |
| IGNORE_VALUE | | 255 | NHEADS | | 8 |
| LOSS_WEIGHT | | 1.0 | CLASS_WEIGHT | | 4.0 |
| CONVS_DIM | | 256 | MASK_WEIGHT | | 5.0 |
| MASK_DIM | ENCODER | 256 | DICE_WEIGHT | DECODER | 5.0 |
| COMMON_STRIDE | | 4 | BOX_WEIGHT | | 5.0 |
| TRANSFORMER_ENC_LAYERS | | 6 | GIOU_WEIGHT | | 2.0 |
| TOTAL_NUM_FEATURE_LEVELS | | 4 | HIDDEN_DIM | | 256 |
| NUM_FEATURE_LEVELS | | 3 | NUM_OBJECT_QUERIES | | 300 |

model approach for testing, *i.e.*, open-vocabulary detection (OVD) + open-vocabulary segmentation (OVS). Here, open-vocabulary detection models contain YOLO-world (Cheng et al., 2024). Besides, the open-vocabulary segmentation model adopts the Clipseg (Lüddecke & Ecker, 2022) to complete the experiments. Note that for the latter (open-vocabulary segmentation model), we initially intended to adopt the SAM model (Kirillov et al., 2023). However, we find that the original SAM requires point or bounding box information for object segmentation. Therefore, in these experiments, we employed the Clipseg model for the open-vocabulary segmentation subtask. The specific experiment results are as shown in Table 13.

Table 12: The proposed OVID task analyses in different open-vocabulary works.

| Name | Instance segmentation work | Panoptic segmentation work |
|---|---|---|
| Does it segment stuff (road/other) | ✗ | ✓ |
| Are output pixels fully covered? | ✗ | ✓ |
| Can it meet the requirements of the OVID task? | ✗ | ✓ |

Table 13: The comparison results between different combined models and our OVIDNet model.

| Model | Type | Domain | Acc | Parameter |
|---|---|---|---|---|
| YOLO-World-S (Cheng et al., 2024)+Clipseg (Lüddecke & Ecker, 2022) | OVD+OVS | Normal | 22.64 | 163.75M |
| | | Foggy | 17.75 | |
| YOLO-World-L (Cheng et al., 2024)+Clipseg (Lüddecke & Ecker, 2022) | OVD+OVS | Normal | 30.08 | 198.75M |
| | | Foggy | 24.34 | |
| OVIDNet(Ours) | End-to-End | Normal | **32.79** | **120.32M** |
| | | Foggy | **27.83** | |

We can find that, compared with the combined model (YOLO-World-L+Clipseg), our model can surpass it by **2.71%** (normal domain) and **3.49%** (foggy domain). These performance gains demonstrate the effectiveness of our model.

## E  MORE RESULTS FOR OVID TASK

In this section, we will provide additional results to test the effectiveness of our framework and strategies. We first report quantitative results of different categories in normal and cross-domain conditions. Then, we present more visualization results. The specific results are shown below.

### E.1  QUANTITATIVE RESULTS OF DIFFERENT CATEGORIES

We first present additional results from various categories using the proposed strategies. Note that we give two types of metrics, *i.e.*, segmentation/detection metrics (IOU, AP, AP@.5) and intrusion

detection metrics (AccY, AccN, Acc), respectively. The former is obtained via a zero-shot manner, the latter via a task-specific transfer manner. We conduct experiments in COCO, Cityscape, and Cityintrusion-OpenV, as shown in Table 14. From Table 14, we can find that when the proposed methodology is added, multiple metrics in multiple categories are improved to a certain extent. Compared with the baseline model, our strategies can surpass it by 3.97% (IOU) and 0.93% (AP), respectively, which verifies the effectiveness of the proposed strategies.

Table 14: The more quantitative results of different intrusion categories. Task: **COCO→Cityscape, Cityintrusion-OpenV**. We provide quantitative results for all possible intrusion categories to test the effectiveness of the proposed strategies. Besides, to comprehensively measure the results across different categories, we report two distinct metrics, *i.e.*, segmentation/detection metrics (IOU, AP, AP@.5) and intrusion detection metrics (AccY, AccN, Acc), respectively. The **bold** is the best result.

| - | | | Intrusion Categories, Task: COCO→Cityscape, Cityintrusion-OpenV | | | | | | | | |
|---|---|---|---|---|---|---|---|---|---|---|---|
| Strategies | | | Segmentation and Detection Metrics | | | | | | | | |
| Baseline | DMG | MDNM | Person(%) | Rider(%) | Car(%) | Truck(%) | Bus(%) | Train(%) | Motorcycle(%) | Bicycle(%) | Avg(%) |
| | | | IOU\|AP\|AP@.5 | IOU\|AP\|AP@.5 | IOU\|AP\|AP@.5 | IOU\|AP\|AP@.5 | IOU\|AP\|AP@.5 | IOU\|AP\|AP@.5 | IOU\|AP\|AP@.5 | IOU\|AP\|AP@.5 | IOU\|AP\|AP@.5 |
| ✓ | ✗ | ✗ | 64.7\|9.1\|23.8 | 0.0\|0.0\|0.0 | 80.8\|17.2\|36.7 | 24.0\|18.6\|25.3 | 62.8\|36.1\|53.9 | 2.2\|13.7\|25.4 | 45.8\|10.3\|25.1 | 69.6\|10.4\|30.4 | 43.74\|14.43\|27.58 |
| ✓ | ✗ | ✓ | 67.2\|10.6\|26.1 | 0.0\|0.0\|0.0 | 82.5\|17.1\|38.7 | 36.0\|19.1\|29.4 | 47.8\|32.4\|52.8 | 4.1\|17.7\|31.8 | 50.9\|8.2\|21.3 | 68.5\|8.8\|28.4 | 44.63\|14.24\|28.56 |
| ✓ | ✓ | ✗ | 69.9\|12.3\|32.1 | 0.0\|0.0\|0.0 | 80.1\|21.4\|45.5 | 24.2\|14.5\|20.5 | 51.0\|36.2\|53.1 | 0.6\|12.5\|20.0 | 56.1\|10.7\|25.7 | 72.2\|11.5\|36.4 | 44.26\|14.89\|**29.16** |
| ✓ | ✓ | ✓ | 66.5\|11.0\|28.6 | 0.0\|0.0\|0.0 | 86.5\|22.8\|47.0 | 37.4\|22.1\|32.9 | 60.2\|33.3\|45.4 | 0.0\|9.6\|13.6 | 58.6\|11.9\|27.8 | 72.5\|12.2\|36.5 | **47.71**\|**15.36**\|28.98 |
| Strategies | | | Intrusion Detection Metrics | | | | | | | | |
| Baseline | DMG | MDNM | Person(%) | Rider(%) | Car(%) | Truck(%) | Bus(%) | Train(%) | Motorcycle(%) | Bicycle(%) | All Categories |
| | | | AccY\|AccN\|Acc | AccY\|AccN\|Acc | AccY\|AccN\|Acc | AccY\|AccN\|Acc | AccY\|AccN\|Acc | AccY\|AccN\|Acc | AccY\|AccN\|Acc | AccY\|AccN\|Acc | AccY\|AccN\|Acc |
| ✓ | ✗ | ✗ | 12.43\|46.39\|39.28 | 0.00\|0.00\|0.00 | 22.74\|29.48\|25.88 | 30.77\|7.32\|20.43 | 31.34\|35.48\|32.65 | 0.00\|0.00\|0.00 | 14.55\|21.28\|18.79 | 12.19\|38.70\|30.55 | 18.72\|36.19\|29.36 |
| ✓ | ✓ | ✗ | 18.02\|42.47\|37.35 | 0.00\|0.00\|0.00 | 22.50\|36.38\|28.97 | 28.85\|4.88\|18.28 | 25.37\|38.71\|29.59 | 0.00\|0.00\|0.00 | 20.00\|26.60\|24.16 | 18.01\|43.12\|35.40 | 20.06\|37.56\|**30.72** |
| ✓ | ✗ | ✓ | 21.51\|39.99\|36.12 | 0.00\|0.00\|0.00 | 22.70\|42.87\|32.10 | 21.15\|14.63\|18.28 | 35.82\|32.26\|34.69 | 11.11\|7.14\|8.70 | 12.73\|23.40\|19.46 | 20.22\|42.26\|35.49 | 21.01\|38.64\|**31.75** |
| ✓ | ✓ | ✓ | 19.55\|41.10\|36.59 | 0.00\|0.00\|0.00 | 28.12\|40.16\|33.73 | 32.69\|7.32\|21.51 | 32.84\|29.03\|31.63 | 0.00\|0.00\|0.00 | 25.45\|15.96\|19.46 | 21.61\|43.61\|36.85 | 24.43\|38.16\|**32.79** |

Table 15: The more quantitative results of different intrusion categories in the cross-domain task. We further test the effectiveness of the proposed strategies with a task-specific transfer manner. Task: **COCO→Foggy-Cityscape, Cityintrusion-OpenV**. Three foggy conditions are used to conduct comprehensive experiments, *i.e.*, $\alpha$=0.005, $\alpha$=0.01, $\alpha$=0.02. The **bold** is the best result.

| - | | | Intrusion Categories, Task: COCO→Foggy-Cityscape, Cityintrusion-OpenV | | | | | | | | |
|---|---|---|---|---|---|---|---|---|---|---|---|
| Strategies | | | $\alpha = 0.005$ | | | | | | | | |
| Baseline | DMG | MDNM | Person(%) | Rider(%) | Car(%) | Truck(%) | Bus(%) | Train(%) | Motorcycle(%) | Bicycle(%) | All Categories |
| | | | AccY\|AccN\|Acc | AccY\|AccN\|Acc | AccY\|AccN\|Acc | AccY\|AccN\|Acc | AccY\|AccN\|Acc | AccY\|AccN\|Acc | AccY\|AccN\|Acc | AccY\|AccN\|Acc | AccY\|AccN\|Acc |
| ✓ | ✗ | ✗ | 14.11\|45.14\|38.64 | 0.00\|0.00\|0.00 | 25.43\|24.38\|24.94 | 28.85\|4.88\|18.28 | 31.34\|29.03\|30.61 | 0.00\|0.00\|0.00 | 9.09\|20.21\|16.11 | 10.25\|37.10\|28.85 | 20.43\|33.58\|28.44 |
| ✓ | ✓ | ✗ | 19.13\|41.21\|36.59 | 0.00\|0.00\|0.00 | 23.67\|30.96\|27.06 | 38.46\|4.88\|23.66 | 29.85\|32.26\|30.61 | 0.00\|0.00\|0.00 | 23.64\|22.34\|22.82 | 18.56\|41.15\|34.21 | 21.29\|34.75\|**29.49** |
| ✓ | ✗ | ✓ | 20.39\|38.18\|34.45 | 0.00\|0.00\|0.00 | 24.07\|38.68\|30.88 | 26.92\|12.20\|20.43 | 31.34\|29.03\|30.61 | 0.00\|7.14\|4.35 | 14.55\|27.66\|22.82 | 20.22\|40.66\|34.38 | 21.66\|36.20\|**30.52** |
| ✓ | ✓ | ✓ | 20.81\|39.55\|35.62 | 0.00\|0.00\|0.00 | 28.28\|35.05\|31.43 | 32.69\|7.32\|21.51 | 38.81\|25.81\|34.69 | 0.00\|0.00\|0.00 | 27.27\|19.15\|22.15 | 22.44\|42.26\|36.17 | 24.96\|35.54\|**31.40** |
| Strategies | | | $\alpha = 0.01$ | | | | | | | | |
| Baseline | DMG | MDNM | Person(%) | Rider(%) | Car(%) | Truck(%) | Bus(%) | Train(%) | Motorcycle(%) | Bicycle(%) | All Categories |
| | | | AccY\|AccN\|Acc | AccY\|AccN\|Acc | AccY\|AccN\|Acc | AccY\|AccN\|Acc | AccY\|AccN\|Acc | AccY\|AccN\|Acc | AccY\|AccN\|Acc | AccY\|AccN\|Acc | AccY\|AccN\|Acc |
| ✓ | ✗ | ✗ | 14.53\|41.66\|35.98 | 0.00\|0.00\|0.00 | 26.11\|20.84\|23.66 | 30.77\|4.88\|19.35 | 29.85\|19.35\|26.53 | 0.00\|0.00\|0.00 | 14.55\|20.21\|18.12 | 13.30\|36.12\|29.11 | 21.29\|30.64\|26.98 |
| ✓ | ✓ | ✗ | 19.55\|38.81\|34.78 | 0.00\|0.00\|0.00 | 24.79\|27.51\|26.06 | 36.54\|4.88\|22.58 | 29.85\|22.58\|27.55 | 0.00\|0.00\|0.00 | 20.00\|22.34\|21.48 | 20.22\|38.94\|33.19 | 22.14\|32.16\|**28.24** |
| ✓ | ✗ | ✓ | 21.09\|36.81\|33.52 | 0.00\|0.00\|0.00 | 25.31\|33.53\|29.14 | 28.85\|7.32\|19.35 | 35.82\|29.03\|33.67 | 11.11\|7.14\|8.70 | 18.18\|27.66\|24.16 | 22.99\|39.07\|34.13 | 23.00\|33.56\|**29.43** |
| ✓ | ✓ | ✓ | 20.95\|37.81\|34.28 | 0.00\|0.00\|0.00 | 30.00\|30.50\|30.23 | 28.85\|7.32\|19.35 | 29.85\|25.81\|28.57 | 0.00\|0.00\|0.00 | 25.45\|20.21\|22.15 | 23.55\|40.17\|35.06 | 25.94\|32.93\|**30.20** |
| Strategies | | | $\alpha = 0.02$ | | | | | | | | |
| Baseline | DMG | MDNM | Person(%) | Rider(%) | Car(%) | Truck(%) | Bus(%) | Train(%) | Motorcycle(%) | Bicycle(%) | All Categories |
| | | | AccY\|AccN\|Acc | AccY\|AccN\|Acc | AccY\|AccN\|Acc | AccY\|AccN\|Acc | AccY\|AccN\|Acc | AccY\|AccN\|Acc | AccY\|AccN\|Acc | AccY\|AccN\|Acc | AccY\|AccN\|Acc |
| ✓ | ✗ | ✗ | 14.53\|37.22\|32.47 | 0.00\|0.00\|0.00 | 26.96\|14.86\|21.32 | 30.77\|2.44\|18.28 | 28.36\|12.90\|23.47 | 0.00\|0.00\|0.00 | 20.00\|20.21\|20.13 | 15.24\|30.96\|26.13 | 22.04\|25.88\|24.38 |
| ✓ | ✓ | ✗ | 21.23\|34.48\|31.71 | 0.00\|0.00\|0.00 | 26.75\|20.10\|23.66 | 32.69\|4.88\|20.43 | 29.85\|16.13\|25.51 | 0.00\|0.00\|0.00 | 23.64\|24.47\|24.16 | 22.44\|33.05\|29.79 | 23.88\|26.90\|**25.72** |
| ✓ | ✗ | ✓ | 23.60\|33.07\|31.09 | 0.00\|0.00\|0.00 | 27.96\|26.36\|27.21 | 30.77\|2.44\|18.28 | 35.82\|19.35\|30.61 | 11.11\|7.14\|8.70 | 16.36\|20.21\|18.79 | 27.42\|33.54\|31.66 | 25.51\|28.50\|**27.33** |
| ✓ | ✓ | ✓ | 21.79\|33.59\|31.12 | 0.00\|0.00\|0.00 | 32.25\|23.83\|28.33 | 28.85\|4.88\|18.28 | 32.84\|16.13\|27.55 | 0.00\|0.00\|0.00 | 25.45\|17.02\|20.13 | 25.48\|34.52\|31.74 | 27.72\|27.90\|**27.83** |

## E.2  MORE RESULTS OF CROSS-DOMAIN TASK

Furthermore, we present additional intrusion detection results for various intrusion categories across different cross-domain tasks. In this experiment, we adopt three different foggy coefficients, *i.e.*, $\alpha$=0.005, $\alpha$=0.01, and $\alpha$=0.02. In these experiments, We conduct experiments in COCO, Foggy-Cityscape, and Cityintrusion-OpenV, as shown in Table 15. We can observe that, in various cross-domain tasks, our strategies enhance intrusion detection performance. In different tasks, compared

with the original baseline model, our framework can improve them by 2.96%, 3.22%, and 3.45%, respectively. Furthermore, our proposed approach can effectively improve the performance of intrusion detection for various categories. These performance improvements demonstrate the effectiveness of our approach, as well as the ability of our framework to generalize.

### E.3 MORE VISUALIZATION COMPARISON RESULTS

Finally, we also present more visualization comparison results to verify the effectiveness of the proposed framework and strategies. We set the text prompt of stuff_classes as Road and set the text prompt of thing_classes as 'Person', 'Rider', 'Car', 'Truck', 'Bus', 'Train', 'Motorcycle', 'Bicycle', as shown in Figure 10. From Figure 10, we can find that our framework can present promising visualization detection results, not only detecting all intruders correctly but also giving correct Intrusion ('Y')/No-intrusion ('N') labels, which proves the effectiveness of our approach.

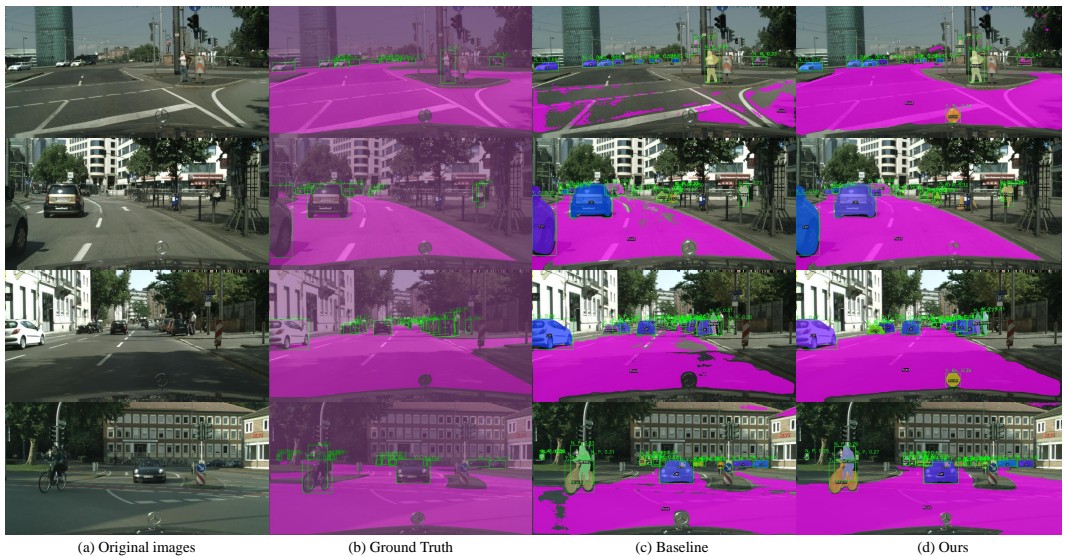

| (a) Original images | (b) Ground Truth | (c) Baseline | (d) Ours |

Figure 10: The visualization comprising results. Here, (a), (b), (c), and (d) denote Original images, Ground truth, Baseline results, and Ours, respectively. Text prompt: Road (stuff_classes), 'Person', 'Rider', 'Car', 'Truck', 'Bus', 'Train', 'Motorcycle', 'Bicycle', (thing_classes). For thing_classes, the abbreviations are used instead of complete labels to easily show our results. The correspondence between the abbreviation and full name can be referred to in the Table 9.

## F THE PRINCIPLES OF GEOMETRIC CONSTRAINT RECLASSIFICATION STRATEGY

In subsection 5.4, we analyze why the performance of the category 'Rider' is '0.0' in depth. To compensate for this gap and enhance the practicality of our model, we design a simple yet effective reasoning enhancement strategy, *i.e.*, Geometric Constraint Reclassification Strategy. Our GCRS strategy is implemented within the post-processing reasoning phase, specifically designed to mitigate local class ambiguity, *e.g.*, category similarity. The ambiguity occurs when our model predicts the person in the vehicle as a standalone Subject Box $B_P$ (Person), rather than the intended composite Rider class. Therefore, the principle of our GCRS strategy relies on utilizing the spatial topological constraints between predicted bounding boxes. Specifically, by calculating the IoU between the detected $B_P$ and any potential Accessory Box $B_V$ (Bicycle/Motorcycle), we quantify their degree of spatial coupling. If this coupling exceeds a predefined High-Coupling Threshold $\tau$, it constitutes a strong geometric constraint. Then, a mandatory class is reassigned to enforce semantic consistency. The core correction criterion is formally expressed as a conditional reclassification operation, and we can express it as

$$\text{If Class}(B_P) = \text{'Person' and } \left(\exists B_V \in \{(\text{'Bicycle', 'Motorcycle'})\} \text{ s.t. IoU}(B_P, B_V) > \tau\right), \\ \text{Then Class}(B_P) \leftarrow \text{'Rider'}, \tag{13}$$

where $\tau$ is a threshold. The IoU metric used to quantify the overlap is defined as

$$\text{IoU}(B_P, B_V) = \frac{\text{Area}(B_P \cap B_V)}{\text{Area}(B_P \cup B_V)}. \tag{14}$$

Our GCRS strategy acts as a domain-knowledge-driven geometric filter. And GCRS can significantly enhance our model's accuracy on some challenging composite instances (*e.g.*, Rider category) by prioritizing geometric evidence over primary model scores.

## G    THE DETAILED INFORMATION OF CITYINTRUSION-OPENV-BDD DATASET

To consider more distribution shift types and enhance the diversity of intrusion scene in open-world deployment, we created a new intrusion detection dataset for the OVID task, namely Cityintrusion-OpenV-BDD. The new dataset is built based on the BDD-100K datasets (Yu et al., 2020). We clean the original dataset based on the proposed OVID task features. The detailed method can refer to Appendix A.1 in our paper. Our new datasets contain rich intrusion scene types, e.g., multiple different weather (Clear, Cloudy, Rainy, Foggy, Night), different geographic environments (City, Highway, Suburban/Rural), different period of time (Daytime, Dusk, Night), and Different transportation environments (Heavy Traffic, Empty Road). These new domains can meet the experiment's requirements in different distribution shifts. Finally, our datasets contain 1482 training data and 449 evaluation data. We evaluate the performance of our model on these datasets, as shown in the Table 7. We can find that, in different domain shifts, our strategies still present performance improvements. Compared with the baseline model, our model can surpass it by **4.58%**, which verifies the strong robustness of our model and the effectiveness of the proposed strategies.

## H    LIMITATIONS AND FURTHER WORKS

In this paper, we introduce the Open-Vocabulary Intrusion Detection (OVID) project for the first time, including a new task, an efficient framework, and a strong benchmark for vision-based intrusion detection. Additionally, we design corresponding strategies to enhance intrusion detection performance in real-world scenarios and increase the practicality of the model. However, there are still some limitations that need to be addressed in the future: (1) Enhance the ability to recognize fine-grained categories and improve generalization performance in the real world. (2) Inspired by methods for parameter-efficient fine-tuning (Hu et al., 2022; Ye et al., 2026), we will explore efficient ways to enhance the intrusion detection performance of models at a lower fine-tuning cost.

