# OpenReview forum: "OVID: Open-Vocabulary Intrusion Detection"
_ICLR.cc/2026/Conference — ICLR 2026 Poster_

### Official Review · Reviewer_aUMD · 2025-10-28

**Soundness:** 2
**Presentation:** 3
**Contribution:** 3
**Rating:** 6
**Confidence:** 4

**Summary:**

The paper introduces OVID, an “open-vocabulary intrusion detection” setting, a new dataset (Cityintrusion-OpenV) built from Cityscapes with 8 intruder classes plus AoI masks, and a baseline model OVIDNet built on OpenSeeD with two add-ons: Multi-Distributed Noise Mixing (MDNM) for box jitter and a Dynamic Memory-Gated (DMG) module. Results show modest gains over OpenSeeD and prior intrusion systems, plus qualitative demos.

**Strengths:**

S1. Problem formulation: Open-vocabulary intrusion detection as a two-stage process: first produce text-conditioned boxes and masks, then make an intrusion decision by measuring overlap with the area of interest. This separation makes the approach easy to follow.

S2. Dataset contribution: Cityintrusion-OpenV extends prior intrusion datasets to eight intruder categories and pairs every image with area-of-interest masks and text prompts. It reports about eighteen interactions per image and gives class-wise counts. This supports both open-vocabulary evaluation and the intrusion decision.

S3. Plug-and-play memory module: The Dynamic Memory-Gated block retrieves a small set of scene features from a fixed memory and fuses them with the decoder through a learned gate. It is a lightweight add-on that aims to capture repeated road layouts and common spatial patterns in driving scenes. The module can be dropped into similar transformer decoders without major changes.

S4. Multi-Distributed Noise Mixing perturbs boxes rather than images. It mixes uniform, gaussian, and Laplace jitters with weights and scales the jitter by box area using a log-like schedule. Small boxes get fine perturbations, large boxes get broader jitter. This is well aligned with an intrusion task where a few pixels of overlap can flip the decision.

**Weaknesses:**

W1. The zero-shot part is mainly domain shift (COCO→Cityscapes). The novel-class aspect is unclear because most classes (person, car, bus, truck, train, motorcycle, bicycle) appear in COCO. Provide a base/novel split where some intruder classes are withheld during training and evaluated only via text prompts (standard OVD protocol). Compare to strong OVD baselines under the same split. (Table 2, Table 10 show no explicit novel-class setup.)

W2. The intrusion decision uses overlap > 20 pixels (fixed) following Sun et al. (Appendix A.1; Sec. 4.2), independent of image scale (800 vs 1200, Table 9 p. 17). This is confusing to me. Report analyses with a normalized overlap (e.g., fraction of box pixels inside AoI) and calibration curves/ROC, and show sensitivity to the threshold t.

W3. The paper argues that OVD+OVS pipelines are costly (Appendix A.5 Table 8 p. 16) but does not report numbers. Please include GroundingDINO(+CLIP) + panoptic/road segmenter (e.g., OpenSeeD/SAM-family) as a task baseline, and a pure OpenSeeD baseline with the exact same training/compute. Also test YOLO-World style detectors with AoI overlap. This is necessary to show OVIDNet improves over obvious modular alternatives.

W4. Improvements over OpenSeeD are ~+2.19 PQ and +3.43 Acc (Table 2 p. 7), and category-wise results contain zeros (e.g., Rider = 0 in several tables; Fig. 6 p. 9). The practical impact of MDNM/DMG is seems modest; please include statistical variance (≥3 seeds) and report confidence intervals, if you have it.

W5. Because of GPU limits, training uses 15k iters and 800px images (Table 9 p. 17). This deviates from OpenSeeD’s stronger recipe (368,750 iters, 1200px). Comparisons must either match schedules/crops or adjust for capacity (e.g., stronger OpenSeeD backbones).

W6. Many visuals imply Road is the AoI (Fig. 5 p. 8; Fig. 10 p. 18), but practical systems often include sidewalks/rails or multiple AoIs. Please clarify whether stuff classes beyond Road are supported and evaluate multi-AoI cases.

W7. Why selectively synthetic fog is only evaluated? For open-world deployment, other distribution-shifts such as common corruptions [1] (rain, snow, so on adapted to Cityscapes) and viewpoint change and perspective distortion [2] (Möbius/MPD augmentation). The scope and reasoning should be clearly defined in paper. Such investigation on robustness of AoI–box consistency under viewpoint change and artifacts can provide worthful insights.
[1] - Hendrycks, Dan, and Thomas Dietterich. "Benchmarking neural network robustness to common corruptions and perturbations." arXiv preprint arXiv:1903.12261 (2019).
[2] - Chhipa, P. C., Chippa, M. S., De, K., Saini, R., Liwicki, M., & Shah, M. (2024, September). Möbius transform for mitigating perspective distortions in representation learning. In European Conference on Computer Vision (pp. 345-363).

**Questions:**

Besides mentioned weakness, following are few questions for authors -
Q1. What are the exact novel classes (if any) absent from training in your zero-shot claims?
Q2. How is class text prompted at test time—single label (“bus”) or templates (“a photo of a bus”)?
Q3. Did you try class re-weighting or few-shot prompts to fix Rider?

---

> ### Author Response · Authors · 2025-11-26
> **The Detailed response for Reviewer aUMD**
>
> **Rebuttal Summary**: We sincerely thank the reviewer for their valuable efforts and constructive feedback. We appreciate the reviewer' recognition of our work's contributions, including: (1) **introduces** OVID, an “open-vocabulary intrusion detection” setting, (2) a **new** dataset (Cityintrusion-OpenV) built and **supports** both open-vocabulary evaluation and the intrusion decision, (3) **easy** to follow the approach, (4) **plug-and-play** memory module, (5) **well aligned** with an intrusion task.
>
> We provided detailed point-by-point responses addressing your concerns below. If you have any further questions, we would be glad to discuss them. Thank you very much.
>
> **-----Weaknesses section response-----**
>
> **W1: The zero-shot part is mainly domain shift (COCO→Cityscapes),......,Table 2, Table 10 show no explicit novel-class setup.**
>
> **A1**: Thanks! In our paper, we report the zero-shot performance in COCO→Cityscapes setting. Our zero-shot performance in Cityscapes is across all categories, not only eight categories, e.g., person, car, bus, truck, train, motorcycle, bicycle. These categories include ‘thing_classes’ and ‘stuff_classes’. Therefore, the metrics we report utilize the Panoptic segmentation metrics RQ(%), SQ(%), and PQ(%). In the cityscape dataset, all categories mainly contain 19 categories, as follows:
>
> [‘road’, ‘sidewalk’, ‘building’, ‘wall’, ‘fence’, ‘pole’, ‘traffic light’, ‘traffic sign’, ‘vegetation’, ‘terrain’, ‘sky’, ‘person’, ‘rider’, ‘car’, ‘truck’, ‘bus’, ‘train’, ‘motorcycle’, ‘bicycle’,]
>
> Besides, our model was trained and tested directly on the Cityscape dataset without any fine-tuning. The setting and writing manner can also be found in some works [1].
>
> **W2: The intrusion decision uses overlap > 20 pixels (fixed) following Sun et al. (Appendix A.1; Sec. 4.2), independent of image scale (800 vs 1200, Table 9 p. 17). This is confusing to me. Report analyses with a normalized overlap (e.g., fraction of box pixels inside AoI) and calibration curves/ROC, and show sensitivity to the threshold t.**
>
> **A2**: Thanks, please let us explain in detail for you.
>
> * **Default settings**. In the field of intrusion detection, the widely accepted pixel threshold is 20. This is primarily because early definitions [2] and studies on intrusion detection consistently set the original pixel threshold to 20 [3][4]. This value is not empirical but derived from experimentation.
>
> * **Additional experiments**:  We also conduct some sensitivity experiments to the threshold t in a more adverse environment dataset. For detailed information about the dataset, please refer to the answer (A7) for Weakness 7. Here, we set the threshold t as 10, 15, 20, 30, respectively, and the results are shown below. We can find that our proposed strategy and framework are insensitive to pixel thresholds. Under the different pixel values, our performance consistently outperforms the baseline and achieves the best results. Besides, as different strategies are added, intrusion performance continues to improve. We can also see that our best performance can still be improved, e.g., when the pixel is 30. The primary reason is that as the pixels increases, the model's sensitivity to intrusion behavior decreases while its sensitivity to non-intrusion behavior increases. According to the computational method employed [2][4], this will lead to an overall increase in the model's sensitivity to intrusions. Therefore, these sensitivity experiments further verify the robustness and efficacy of the proposed strategy and framework across varying environments and settings.
> | Method     |  | t=10  | |  | t=15  | | | t=20  | |  | t=30  | |
> | ----------- | ----------- | ----------- | ----------- | ----------- | ----------- | ----------- | ----------- | ----------- | ----------- | ----------- | ----------- | ----------- |
> |  - |AccY|AccN|Acc|AccY|AccN |Acc |AccY | AccN |Acc | AccY | AccN |Acc |
> |  Baseline |22.32 |15.74 |18.31 |21.64 | 16.55 | 18.54 |20.99 | 17.22 |18.69 |19.93 | 18.45 |19.03 |
> |  +DMG | 21.86 | 17.30 | 19.09 |20.83 | 18.27 | 19.27 | 20.16 | 19.27 | 19.62 |19.15 | 20.75 |20.13 |
> |  +MDNM | 21.81 | 18.17 |19.60 |20.96 | 19.34 |19.97 |20.26 | 20.38 | 20.33 |19.08 | 22.06 |20.89 |
> |  +DMG+MDNM | 28.30 | 19.01 | 22.64 |26.74 | 20.46 |22.92 |**25.79** | **21.66** |**23.27**|24.45| 23.59 |23.93 |

---

> ### Author Response · Authors · 2025-11-26
>
> **W3: The paper argues that OVD+OVS pipelines are costly (Appendix A.5 Table 8 p. 16) but does not report numbers. Please include GroundingDINO(+CLIP) + panoptic/road segmenter (e.g., OpenSeeD/SAM-family) as a task baseline, and a pure OpenSeeD baseline with the exact same training/compute. Also test YOLO-World style detectors with AoI overlap. This is necessary to show OVIDNet improves over obvious modular alternatives.**
>
> **A3**:  Good question and suggestion! To validate the effectiveness of our proposed model, we conducted some experiments on the YOLO-world and GroundingDINO models. Specifically, we follow previous intrusion detection work [2] [4] and adopt a combined model approach for testing, i.e., open-vocabulary detection (OVD) + open-vocabulary segmentation (OVS). Here, open-vocabulary detection models contain YOLO-world. Besides, the open-vocabulary segmentation model adopts the Clipseg to complete the experiments. Note that for the latter (open-vocabulary segmentation model), we initially intended to adopt the SAM model. However, we find that the original SAM requires point or bounding box information for object segmentation. Therefore, in these experiments, we employed the Clipseg model for the open-vocabulary segmentation subtask. The specific experiment results are as follows:
> | Model | Type | Domain | Acc | Parameter |
> | :---: | :---: | :---: | :---: | :---: |
> | YOLO-World-S [5] +Clipseg [6] | OVD+OVS | Normal | 22.64 | 163.75M |
> | | | Foggy | 17.75 | |
> | YOLO-World-L [5] +Clipseg [6]  | OVD+OVS | Normal | 30.08 | 198.75M |
> | | | Foggy | 24.34 | |
> | **OVIDNet (Ours)** | End-to-End | Normal | **32.79** | **120.32M** |
> | | | Foggy | **27.83** | |
>
> We can find that compared with the combined model ( YOLO-World-L +Clipseg), our model can surpass it by **2.71%** (normal domain) and **3.49%** (foggy domain). These performance gains demonstrate the effectiveness of our model.

---

> ### Author Response · Authors · 2025-11-26
>
> **W4: Improvements over OpenSeeD are ~+2.19 PQ and +3.43 Acc (Table 2 p. 7), and category-wise results contain zeros (e.g., Rider = 0 in several tables; Fig. 6 p. 9). The practical impact of MDNM/DMG is seems modest; please include statistical variance (≥3 seeds) and report confidence intervals, if you have it.**
>
> **A4: (1) sub-weakness1: Rider = 0 in several tables; Fig. 6 p. 9**
>
> In the insightful and interesting experiments (Section 5.4), we conducted an in-depth analysis of this phenomenon and its reasons. To compensate for this gap and enhance the practicality of our model, we design a simple yet effective reasoning enhancement strategy, as follows:
>
> **Geometric Constraint Reclassification Strategy (GCRS)**: The GCRS strategy is implemented within the structured post-processing phase of the object detection pipeline, specifically designed to mitigate local class ambiguity arising from the model's decomposition of compound semantic entities like the Rider, e.g., category similarity. The ambiguity occurs when the model predicts the person on the vehicle as a standalone Subject Box $B_P$ (Person), rather than the intended composite Rider class. The principle of the GCRS strategy relies on utilizing the spatial topological constraints between predicted bounding boxes. By calculating the IoU between the detected $B_P$ and any potential Accessory Box $B_V$ (Bicycle/Motorcycle), we quantify their degree of spatial coupling. If this coupling exceeds a predefined High-Coupling Threshold $\tau$, it constitutes a strong geometric constraint, necessitating a mandatory class reassignment to enforce semantic consistency. The core correction criterion is formally expressed as a conditional reclassification operation:
>
> $$\text{If } \text{Class}(B_P) = \text{'Person'} \text{ and } \left( \exists B_V \in \{(\text{'Bicycle'}, \text{'Motorcycle'})\} \text{ s.t. } \text{IoU}(B_P, B_V) > \tau \right), \text{Then } \text{Class}(B_P) \leftarrow \text{'Rider'}$$
>
> where $\tau$ is a threshold. The IoU metric used to quantify the overlap is defined as:
>
> $$\text{IoU}(B_P, B_V) = \frac{\text{Area}(B_P \cap B_V)}{\text{Area}(B_P \cup B_V)}$$
>
> GCRS strategy acts as a domain-knowledge-driven geometric filter, significantly enhancing the model's accuracy on these challenging composite instances by prioritizing geometric evidence over primary model scores.
>
> **Experiment results**: Finally, we conducted several experiments to verify the effectiveness of the proposed GCRS strategy, as shown below. We can find that when applying the proposed strategy, the performance of the rider category improved from 0 to 23.56. Additionally, intrusion detection performance has improved, i.e., 32.79→33.55. In future research, we will explore more effective strategies to continue improving performance.
>
> | Performance      | Rider_Y |Rider_N |Rider |Acc
> | ----------- | ----------- |----------- |----------- |----------- |
> | OVIDNet (ours)      | 0      | 0 | 0 | 32.79
> | + GCRS   | 11.95      | 31.52 | 23.56 | **33.55**
>
> **(2) sub-weakness2: The practical impact of MDNM/DMG is seems modest; please include statistical variance (≥3 seeds) and report confidence intervals, if you have it.**
>
> Thank you for the constructive suggestion regarding the statistical significance of our results. We fully agree that reporting performance variance is important for a more reliable evaluation. We have conducted three independent runs for all compared methods and report the mean and standard deviation. These detailed results are shown below. We observe that the variance across the three runs is small, indicating that the results are stable and reproducible. This consistency further supports the reliability of our MDNM/DMG strategy.
>
> | Method      | First | Second | Third | Avg+std
> | ----------- | ----------- | ----------- | ----------- |----------- |
> | Baseline      | 29.36       | 29.35  | 29.37  | 29.36±0.01
> | +DMG     | 30.72       | 30.71  |  30.72  | 30.72±0.01
> | +MDNM      | 31.70       | 31.65  | 31.75  | 31.70±0.05
> | + DMG+MDNM      | 32.83       | 32.70  | 32.69  | **32.74±0.08**

---

> ### Author Response · Authors · 2025-11-26
>
> **W5: Because of GPU limits, training uses 15k iters and 800px images (Table 9 p. 17). This deviates from OpenSeeD’s stronger recipe (368,750 iters, 1200px). Comparisons must either match schedules/crops or adjust for capacity (e.g., stronger OpenSeeD backbones).**
>
> **A5**: Good question! To verify the effectiveness of the proposed methods and model, we conduct additional experiments with a higher 1200 resolution and more 368,750 training iterations, as shown in the table below. We can find that not only in Setting 1 ( image\_size=800,  iterations=15000) but also Setting 2 (image\_size=1200,  iterations=368,750), our model performance is still better than the baseline model, which demonstrates the effectiveness of the proposed methods and frameworks.
>
> | Setting      | model  | AccY | AccN | Acc
> | ----------- | ----------- |----------- |----------- |---------- |
> | image\_size=800,  iterations=15000    | baseline   |18.72 | 36.19 | 29.36
> | | Ours      | 24.43 | 38.16 | **32.79**
> | image\_size=1200,  iterations=368,750   |  baseline  | 40.06 |  25.30 | 31.07
> | | Ours    | 45.51 |  24.98 | **33.01**
>
> **W6: Many visuals imply Road is the AoI (Fig. 5 p. 8; Fig. 10 p. 18), but practical systems often include sidewalks/rails or multiple AoIs. Please clarify whether stuff classes beyond Road are supported and evaluate multi-AoI cases.**
>
> **A6**: Thank you for the reviewer’s valuable comment.
>
> * **Generalization evaluation under diverse conditions**: Our work targets autonomous-driving safety scenarios, where Road is the primary and most safety-critical Area of Interest (AoI). As presented in the paper, we have already evaluated the proposed framework under multiple monitoring environments, various adverse-weather conditions, and diverse prompt combinations, demonstrating promising generalization across both visual domains and linguistic variations. It is also important to emphasize that all experiments are conducted in a zero-shot setting, without any fine-tuning on autonomous-driving datasets. Visualized experimental results show that our framework has adaptability and application potential in realistic open-world driving settings.
>
> * **Why sidewalk and railway are challenging in zero-shot settings**:  Since we did not fine-tune on any autonomous driving or road/sidewalks/rails-related datasets, and in our framework, we adopt the clip to extract features. CLIP-pretrained models can segment it because the visual patterns of roads are frequent and visually consistent, allowing the model to learn strong implicit alignment with related textual concepts like “street” or “road.” In contrast, sidewalks and railways appear less frequently and exhibit more diverse appearances, leading to weaker representations in CLIP’s embedding space and making them harder to segment in a zero-shot manner. However, we can use reformulated prompts to match the semantic information between the two, e.g., 'Sidewalks: A Road of Person Walk' --> 'Sidewalks: A Road of Person Walk'. Then, in this way, we can get the final results. We present some visualizations in Appendix E.
>
> * **Future work: enhancing semantic coverage and robust generalization**: In future work, we will further extend our system to fully support multiple AoIs by targeted data enrichment and improved semantic grounding methods. We believe such developments will further enhance the generalizability and applicability of the proposed framework in broader surveillance and transportation scenarios.

---

> ### Author Response · Authors · 2025-11-26
>
> **W7: Why selectively synthetic fog is only evaluated? For open-world deployment,......,and artifacts can provide worthful insights.**
>
> **A7**: Good question! To consider more distribution shift types and enhance the diversity of intrusion scene in open-world deployment, we created a new intrusion detection dataset for the OVID task, namely Cityintrusion-OpenV-BDD. The new dataset is built based on the BDD-100K datasets [7]. We clean the original dataset based on the proposed OVID task features. The detailed method can refer to Appendix A.1 in our paper. Our new datasets contain rich intrusion scene types, e.g., multiple different weather (Clear, Cloudy, Rainy, Foggy, Night), different geographic environments (City, Highway, Suburban/Rural), different period of time (Daytime, Dusk, Night), and Different transportation environments (Heavy Traffic, Empty Road). These new domains can meet the experiment's requirements in different distribution shifts. Finally, our datasets contain 1482 training data and 449 evaluation data. We evaluate the performance of our model on these datasets, as shown in the table below. We can find that, in different domain shifts, our strategies still present performance improvements. Compared with the baseline model, our model can surpass it by 4.58%, which verifies the strong robustness of our model and the effectiveness of the proposed strategies.
> | Baseline | DMG | MDNM | AccY (%) | AccN (%) | Acc (%) | Gain |
> | :---: | :---: | :---: | :---: | :---: | :---: | :---: |
> | $\checkmark$ |$\times$ | $\times$ | 20.99 | 17.22 | 18.69 |- |
> | $\checkmark$ | $\checkmark$ | $\times$ | 20.16 | 19.27 | 19.62 | +0.93 |
> | $\checkmark$ | $\times$ | $\checkmark$ | 20.26 | 20.38 | 20.33 | +1.64 |
> | $\checkmark$ | $\checkmark$ | $\checkmark$ | 25.79 | 21.66 | **23.27** | +4.58 |
>
> **-----Questions section response-----**
>
> **Q1:What are the exact novel classes (if any) absent from training in your zero-shot claims?**
>
> **A1**: Thanks! In the zero-shot claims, multiple categories are novel, with these categories primarily concentrated in stuff classes, we list them as follow:
>
> [road’, ‘sidewalk’, ‘building’, ‘wall’, ‘fence’, ‘pole’, ‘traffic light’, ‘traffic sign’, ‘vegetation’, ‘terrain’, ‘sky’]
>
> More detailed information, please refer the answer of weakness1, thank you.
>
>
> **Q2:How is class text prompted at test time—single label (“bus”) or templates (“a photo of a bus”)?**
>
> **A2**:  Thanks. In the test time, we use a single label to test the performance, as follows.
>
> thing_classes = ['Person', 'Rider', 'Car', 'Truck', 'Bus', 'Train', 'Motorcycle', 'Bicycle']
>
> stuff_classes = ['Road']
>
> **Q3:Did you try class re-weighting or few-shot prompts to fix Rider?**
>
> **A3**: This is a good suggestion to improve the performance of Rider classes. We try to set multiple different prompts to solve the issue. However, performance improvements for the rider category do not appear to be very significant. Fortunately, we designed a simple yet effective reasoning enhancement strategy: **Geometric Constraint Reclassification Strategy (GCRS)** to improve the performance of the rider category. For specific details, please refer to the first answer under Weakness 4. Thanks.
>
> **Reference**
>
> [1] Zhang H, Li F, Zou X, et al. A simple framework for open-vocabulary segmentation and detection[C]//Proceedings of the IEEE/CVF International Conference on Computer Vision. 2023: 1020-1031.
>
> [2] Sun J, Chen J, Chen T, et al. PIDNet: An efficient network for dynamic pedestrian intrusion detection[C]//Proceedings of the 28th ACM International Conference on Multimedia. 2020: 718-726.
>
> [3] Shi Z, He S, Sun J, et al. An efficient multi-task network for pedestrian intrusion detection[J]. IEEE transactions on intelligent vehicles, 2022, 8(1): 649-660.
>
> [4] Han F, Ye P, Li K, et al. Mf-id: a benchmark and approach for multi-category fine-grained intrusion detection[J]. IEEE Transactions on Automation Science and Engineering, 2024, 22: 3582-3597.
>
> [5] Cheng T, Song L, Ge Y, et al. Yolo-world: Real-time open-vocabulary object detection[C]//Proceedings of the IEEE/CVF conference on computer vision and pattern recognition. 2024: 16901-16911.
>
> [6] Lüddecke T, Ecker A. Image segmentation using text and image prompts[C]//Proceedings of the IEEE/CVF conference on computer vision and pattern recognition. 2022: 7086-7096.
>
> [7] Yu F, Chen H, Wang X, et al. Bdd100k: A diverse driving dataset for heterogeneous multitask learning[C]//Proceedings of the IEEE/CVF conference on computer vision and pattern recognition. 2020: 2636-2645.

---

### Official Review · Reviewer_jM2n · 2025-10-31

**Soundness:** 3
**Presentation:** 3
**Contribution:** 3
**Rating:** 6
**Confidence:** 3

**Summary:**

The paper introduces Open-Vocabulary Intrusion Detection (OVID), which defines a new task, provides a Cityscapes-derived dataset (Cityintrusion-OpenV), and proposes a multi-modal, multi-task framework (OVIDNet) that aligns image features with text prompts to jointly perform detection, segmentation, and intrusion judgment. The method uses pixel overlap between predicted boxes and AoI masks to identify intrusions. Two components, Multi-Distributed Noise Mixing (MDNM) for adaptive perturbations and a Dynamic Memory-Gated (DMG) module for context, are designed to improve zero-shot generalization and open-world robustness. Reported gains over OpenSeeD are +2.19 PQ and +3.43 intrusion accuracy under lighter training settings (800 resolution, 15K iterations). The dataset extends prior intrusion benchmarks from 4 to 8 categories with Y/N labels and text prompts (18.03 cases per image). Qualitative results on ShanghaiTech and UA-DETRAC show promptable deployment without retraining, although quantitative metrics are missing for those examples.

**Strengths:**

The paper is the first to formally define open-vocabulary intrusion detection and presents a multi-modal framework that combines detection, segmentation, and intrusion judgment. It connects open-vocabulary detection with security applications in an interesting and practical way.

The Cityintrusion-OpenV dataset expands existing intrusion datasets to 8 categories and introduces text prompts with a much higher number of Y/N cases per image (18.03), which is a solid step toward open-world evaluation.

The experimental coverage is broad, including zero-shot transfer (COCO→Cityscapes), task-specific transfer (Cityintrusion-OpenV), fog domain testing, ablations, and qualitative real-world demonstrations.

The ablations for MDNM and DMG demonstrate how different configurations (α, β, γ values, and memory unit sizes) affect performance and reveal consistent, though modest, improvements.

The figures and algorithm descriptions are clear and easy to follow, and the appendix includes sufficient details for reproduction.

The approach shows potential for practical deployment, as it can handle new categories via prompts without retraining and appears somewhat robust under foggy conditions.

**Weaknesses:**

The baseline comparisons are not entirely fair. The authors trained with a reduced image size (800) and fewer iterations (15K), while OpenSeeD used 1200 and 368,750. Without matching settings or reporting variance over multiple seeds, it is hard to interpret the reported gains.

The ShanghaiTech and UA-DETRAC results are only qualitative, but the paper makes strong claims of universal applicability. Quantitative results would be needed to back that up.

The technical novelty is limited since OVIDNet largely builds on OpenSeeD with an added intrusion decision module. MDNM’s mixture of distributions and logarithmic area scaling lacks theoretical justification, and DMG feels similar to standard attention mechanisms without a clear distinction.

The reported improvements are small (1–3 percent), and in some cases, performance drops, showing that gains are not stable.

The absolute performance remains low (32.79 percent intrusion accuracy), and the Rider class fails completely (0.0 percent) due to confusion with Person and class imbalance. The paper does not address this failure.

Comparisons with other strong open-vocabulary detection systems, such as YOLO-World and Grounding DINO, are missing, which weakens the SOTA claim.

The dataset labeling relies on an automatic 20-pixel overlap rule with limited manual verification. There is no inter-annotator agreement, threshold sensitivity analysis, or bias auditing.

**Questions:**

Please follow Strengths and Weaknesses.

Why is only selectively synthetic fog evaluated? For open-world deployment, it would be important to also consider other distribution shifts such as common corruptions [1] (rain, snow, and similar variations adapted to Cityscapes) and viewpoint or perspective changes [2] (for example, Möbius or MPD augmentations). The paper should clearly define the scope and reasoning behind focusing solely on fog. Exploring how AoI–box consistency behaves under viewpoint changes and visual artifacts could provide valuable insights into the model’s robustness.

[1] - Hendrycks, Dan, and Thomas Dietterich. "Benchmarking neural network robustness to common corruptions and perturbations." arXiv preprint arXiv:1903.12261, 2019.
[2] - Chhipa, P. C., Chippa, M. S., De, K., Saini, R., Liwicki, M., & Shah, M.  Möbius transform for mitigating perspective distortions in representation learning. In European Conference on Computer Vision, 2024.

What happens if the models are trained at higher resolution (1200) and for longer schedules? Do the reported gains hold?



I am open to revising my score.

---

> ### Author Response · Authors · 2025-11-26
> **The Detailed response for Reviewer jM2n**
>
> **Rebuttal Summary**: We sincerely thank the reviewer for their valuable efforts and constructive feedback. We appreciate the reviewer' recognition of our work's contributions, including: (1) the **first** formally define open-vocabulary intrusion detection, (2) a **solid step** toward open-world evaluation, (3) **broad** experimental coverage and ablation results; (4) **clear** figures and algorithm descriptions, **sufficient** details for reproduction, and **easy** to follow, (5) **strong** potential for practical deployment.
>
> We provided detailed point-by-point responses addressing your concerns below. If you have any further questions, we would be glad to discuss them. Thank you very much.
>
> **-----Weaknesses section response-----**
>
> **W1: The baseline comparisons are not entirely fair,......, hard to interpret the reported gains.**
>
> **A1**:  Good question! To verify the effectiveness of the proposed methods and model, we conduct additional experiments with a higher 1200 resolution and more 368,750 training iterations, as shown in the table below. We can find that not only in Setting 1 ( image\_size=800,  iterations=15000) but also Setting 2 (image\_size=1200,  iterations=368,750), our model performance is still better than the baseline model, which demonstrates the effectiveness of the proposed methods and frameworks.
>
> | Setting      | model  | AccY | AccN | Acc
> | ----------- | ----------- |----------- |----------- |---------- |
> | image\_size=800,  iterations=15000    | baseline   |18.72 | 36.19 | 29.36
> | | Ours      | 24.43 | 38.16 | **32.79**
> | image\_size=1200,  iterations=368,750   |  baseline  | 40.06 |  25.30 | 31.07
> | | Ours    | 45.51 |  24.98 | **33.01**
>
> **W2: The ShanghaiTech and UA-DETRAC results,......, needed to back that up.**
>
> **A2**: This is a good question and suggestion! Please allow us to explain why we did not provide quantitative results on these two datasets.
>
> * **Lack of Intrusion Labels**: While writing this paper, we also considered providing detailed quantitative results. However, after conducting a thorough investigation of these two datasets, we find that these two datasets **only detection label** are available, which does not meet the requirements of our OVID.  The main reason is that our OVID task is a multi-task involving **detection, segmentation, and intrusion detection**. Consequently, for these two datasets, we are unable to construct original intrusion and non-intrusion labels, i.e., ground truth. This is one of the primary reasons we cannot provide quantitative results on these two datasets. Regarding this point, we mentioned it in the insightful Experiment 2 in Section 5.4 (lines 458–462).
>
> * **Real-scenario application exploration**: Although we cannot provide detailed quantitative results, we still want to demonstrate the broad applicability and effectiveness of our proposed model and methodology through experimentation. Here, we draw inspiration from work in the super-resolution field to validate this perspective through a visualization experiment [1][2]. We use different domains, e.g., Normal, Rainy, and Night, to evaluate the model’s generalization performance with 2, 3, and 9 prompts, as shown in Figure 7 in our paper. Note that the prompts are customizable. We can observe that our framework can detect and judge intrusion behavior, demonstrating the practicality and effectiveness of the proposed framework.
>
> * **Generalization verification**: To provide more qualitative results on other datasets, we also create a new intrusion detection dataset for the OVID task, namely Cityintrusion-OpenV-BDD. The new dataset is built based on the BDD-100K datasets [3]. For detailed information, please refer to Q1’s response. Our new datasets contain rich intrusion scene types, e.g., multiple different weather (Clear, Cloudy, Rainy, Foggy, Night), different geographic environments (City, Highway, Suburban/Rural), different period of time (Daytime, Dusk, Night), and Different transportation environments (Heavy Traffic, Empty Road). We can find that, in different domain shifts, our strategies still present promising performance improvements, compared with the baseline model, our model can surpass it by **4.58%**, which verifies the strong robustness of our model and the effectiveness of the proposed strategies.
> | Baseline | DMG | MDNM | AccY (%) | AccN (%) | Acc (%) | Gain |
> | :---: | :---: | :---: | :---: | :---: | :---: | :---: |
> | $\checkmark$ |$\times$ | $\times$ | 20.99 | 17.22 | 18.69 |- |
> | $\checkmark$ | $\checkmark$ | $\times$ | 20.16 | 19.27 | 19.62 | +0.93 |
> | $\checkmark$ | $\times$ | $\checkmark$ | 20.26 | 20.38 | 20.33 | +1.64 |
> | $\checkmark$ | $\checkmark$ | $\checkmark$ | 25.79 | 21.66 | **23.27** | +4.58 |
>
> * **Further works**: Additionally, in future work, we will obtain the final intrusion labels through **manual annotation** and conduct further experiments to validate the effectiveness of our approach.

---

> ### Author Response · Authors · 2025-11-26
>
> **W3: The technical novelty is limited since OVIDNet largely builds on OpenSeeD with an added intrusion decision module. MDNM’s mixture of distributions and logarithmic area scaling lacks theoretical justification, and DMG feels similar to standard attention mechanisms without a clear distinction.**
>
> **A3**: Thanks! Regarding the question, we would like to explain it from the following three aspects for you:
>
> **(1) Sub-weakness1: The technical novelty is limited since OVIDNet largely builds on OpenSeeD with an added intrusion decision module.**
>
> For this sub-weakness, we want to highlight and emphasize our contributions and innovations, as follows:
>
> * **Novel task and dataset.** Our main contribution lies in defining and analyzing the new OVID task, which introduces task-aware, context-dependent intrusion reasoning under open-world conditions— *a problem not addressed by prior OVD/OVS works*. To the best of our knowledge, the OVID task is proposed for the first time, and this is the first try in the vision-based intrusion field.
>
> * **Effective design and strategy.** Besides, to accomplish the novel OVID task, we design an effective and multi-modal framework.
> In response to the characteristics of the tasks we have proposed, we have also designed corresponding strategies: (1) A Multi-Distributed Noise Mixing strategy is introduced to enhance the location information of unknown and unseen categories. (2) A Dynamic Memory-Gated module is designed to capture the contextual information under complex scenarios. These two effective strategies can improve the generalization and enhance intrusion detection performance of the framework. Comprehensive experiments and comparisons denote that these designs and strategies are feasible and innovative. Our model is built upon a promising foundation. Achieving a 1-3\% gain on a highly competitive benchmark indicates that our modules are successfully pushing the performance ceiling, which is a substantial contribution.
>
> **(2) Sub-weakness2: Mechanism proof for Multi-Distributed Noise Mixing strategy**
>
> To clarify why the proposed noise mixture improves generalization toward unknown categories, we provide a theoretical proof based on Vicinal Risk Minimization (VRM). In our paper, our perturbation can be written as:
>
> $B_{f}=\mathcal{C}\left [ B_{e}+Z\odot \bigtriangleup \odot \Theta ,\textbf{ 0}, \textbf{ 1} \right ],$
>
> where $Z=\alpha \cdot \textbf{N}{u}+\beta  \cdot \textbf{N}{g} +\gamma \cdot \textbf{N}_{t} $.  And
> $\textbf{N}_u $,
> $\textbf{N}_g $,
> $\textbf{N}_t $ denote noise sampled from Uniform, Gaussian, and Laplace distributions, respectively. Since $Z$ is a combination of three independent noise sources, its distribution can be expressed as:
>
> $p_Z(z) = \alpha p_u(z) + \beta p_g(z) + \gamma p_t(z),$
>
> which is a mixed distribution containing (1) fine-grained bounded perturbations (U), (2) moderate Gaussian variations (Gaussian), (3) heavy-tailed structural deviations (Laplace). This directly yields a mixed vicinal distribution in the Vicinal Risk Minimization (VRM) framework. Following Chapelle et al. [4], the vicinal risk can be written as:
>
> $\hat{R} _ {\text{VRM}}(f) = \mathbb{E} _ {(x,y) \sim \mathcal{D}} \mathbb{E} _ {z \sim \mathcal{ p _ Z }} \ell(f(x+z), y).$
>
> Because $\mathcal{ p_Z }$ is a mixture, VRM can be expressed as:
>
> $\hat{R}\_{\text{M-VRM}}(f) = \alpha \hat{R}_u(f) + \beta \hat{R}_g(f) + \gamma \hat{R}_t(f),$
>
> where $\hat{R} _ {u}(f) = \mathbb{E} _ {z \sim p _ u} \ell (f(x+z), y), \hat{R} _ {g}(f) = \mathbb{E} _ {z \sim p_g} \ell (f(x+z), y), \hat{R} _ {t}(f) = \mathbb{E} _ {z \sim p_t} \ell (f(x+z), y).$ Thus, instead of training against a single perturbation model, our model effectively optimizes risk under three complementary vicinal neighborhoods simultaneously.
>
> Let the real open-world perturbation distribution be $q(z)$. If $q$ and $p_Z$ share support and $q \ll p_Z$, then the Radon-Nikodym ratio can be expressed as:
>
> $$K := \sup_z \frac{q(z)}{p_Z(z)},$$
>
> and the $K$ is finite, giving:
>
> $$q(z) \le K p_Z(z).$$
>
> Because $p_Z(z) = \alpha p_u(z) + \beta p_g(z) + \gamma p_t(z)$ covers bounded signals (U), smooth variations (Gaussian), and heavy-tailed structural changes (Laplace). It has wider support and a smaller worst-case ratio $𝐾$ than any single distribution. Therefore, the true risk is upper-bounded by:
>
> $ R_q(f) =  \mathbb{E} _ {z \sim q} \ell (f(x+z), y) \le K \hat{R} _ {\text{M-VRM}}(f). $
>
> A smaller constant $𝐾$ implies a tighter bound. Hence, our model can better generalize to unknown object shapes, sizes, and distributions, and improve localization for novel or rare categories. Because their perturbations are more likely to fall inside the large support of the mixture $p_Z$.

---

> ### Author Response · Authors · 2025-11-26
>
> **(3) Sub-weakness3: Mechanism Explanation for Dynamic Memory-Gated Module**
>
> **(i)  The Core Distinction: Dual-Path Architecture and Information Source**:  Our DMG module is fundamentally different from standard attention in its dual-path architecture and fusion strategy, as detailed below:
>
> Standard attention mechanisms (e.g., Self-Attention or Scaled Dot-Product Attention) operate on a single stream to re-weight input features. In contrast, DMG is a dual-path system that processes information from two distinct sources before fusion:
>
> * **Path 1: External Context Retrieval (Memory Networks)**. This path, represented by Equation (4), is a Memory-Augmented attention mechanism. It computes the retrieval output $O_m$ by matching the input query $Q$ (global context vector) against an External Memory Bank ($M_K$ and $M_V$, which are learnable, decoupled knowledge bases). Unlike standard Self-Attention where $K$ and $V$ are derived directly from the input feature $X$, our $M_K$ and $M_V$ are fixed, global memory units (with dimension $M \times C$) that store generalized, long-term contextual knowledge. This retrieval of external, non-local information is the first major deviation from conventional mechanisms.
>
> * **Path 2: External Context Retrieval (Memory Networks)**.  This path generates the dynamic weight vector $W = \sigma(W_2 \text{ReLU}(W_1 Q))$. This weight is generated by a two-layer MLP (Fully Connected networks $F1, R, F2, S$ in the diagram) and is applied as a dynamic gate. This gate is calculated independently of the memory retrieval process and serves to adaptively adjust the magnitude and distribution of the input feature $X$ via a Hadamard product (⊗) to generate the adjusted features ($X \odot W$). This step introduces a non-linear, adaptive control layer over the primary feature stream.
>
> **(ii) Distinction in Feature Fusion Strategy (Equation 5)**: The final step of the DMG module, detailed in Equation (5), represents the most significant structural departure from standard attention, which typically relies on simple element-wise addition or multiplication:
>
> $$X_f = \text{Conv}1\times 1(\text{Concat}(X \odot W), O_m),$$
>
> * **Standard Attention Fusion**: Usually employs Residual Connection ($X_{out} = X_{in} + \text{Attention}(X_{in})$) or a simple element-wise multiplication.
>
> * **DMG Fusion**: Fuses the two streams using Concatenation followed by a $1\times 1$ Convolution. The two inputs to the concatenation are: (1) The dynamically gated input features ($X \odot W$); and (2) The retrieved external memory output ($O_m$). This Concatenation $\to 1\times 1$ Conv sequence allows the model to learn a complex, non-linear mixture of the refined local features and the retrieved global context, enabling richer representational learning than simple additive or multiplicative scaling.
>
> **(iii) Empirical Demonstration of Significance**: We demonstrate the significance of the DMG module in our ablation study (Table 5). Specifically, the DMG module is vital for improving performance on the challenging Intrusion Judgment metric. Removing the DMG leads to a substantial decrease of 1.36% in the final detection and judgment accuracy.
>
> | Baseline       | Memeroy units |  Acc(%) |
> | ----------- | ----------- |----------- |
> | √      | ×      | 29.36 |
> | √    | √ (M=30) | 30.31 |
> | √    | √ (M=40) | **30.72** |
> | √    | √ (M=50) | 28.42 |
>
> **(iv) Validation on Adverse Environment Datasets**:  To verify the effectiveness of the proposed DMG strategy, we also create a new intrusion detection dataset for the OVID task, namely Cityintrusion-OpenV-BDD. The new dataset is built based on the BDD-100K datasets. Our new datasets contain rich intrusion scene types, e.g., multiple different weather (Clear, Cloudy, Rainy, Foggy, Night), different geographic environments (City, Highway, Suburban/Rural), different period of time (Daytime, Dusk, Night), and Different transportation environments (Heavy Traffic, Empty Road). We can find that, in different domain shifts, compared with the baseline model, our model can surpass it by 0.93%. These performance improvements denote that our DMG strategy is effective, especially in facing and dealing with novel and adverse intrusion scenarios.
>
> | Baseline       | Memeroy units |  Acc(%) |
> | ----------- | ----------- |----------- |
> | √      | ×      | 18.69 |
> | √      | √ (M=40)      | **19.62** |

---

> ### Author Response · Authors · 2025-11-26
>
> **W4: The reported improvements are small (1–3 percent), and in some cases, performance drops, showing that gains are not stable.**
>
> **A4**: Thanks! Let us explain for you:
>
> (1) **sub-weakness1: The reported improvements are small (1–3 percent)**
>
> **Significance of Improvement in OVID Context**:  While the absolute improvements in overall average metrics (e.g., Intrusion  Acc) may appear to be in the 1-3 percent range, this margin is highly significant, especially when built upon a strong baseline like the OpenSeeD architecture.
>
> * **Our contribution**: Our model is built upon a promising foundation. Achieving a $1-3\%$ gain on a highly competitive benchmark indicates that our modules are successfully pushing the performance ceiling, which is a substantial contribution.
>
> * **Targeted Improvement**: The primary purpose of the DMG and MDNM modules is not to maximize easy gains, but to enhance robustness and generalization on the most challenging subsets of the OVID task: (1) the Multi-Distributed Noise Mixing strategy is introduced to enhance location information of unknown and unseen categories, (2) the Dynamic Memory Gated module is designed to capture the contextual information under complex scenarios.
>
> (2) **sub-weakness2: in some cases, performance drops, showing that gains are not stable**
>
> Regarding this issue, we give the following explanation and analysis:
>
> * **Zero-Shot Generalization Behavior**: Because our experiments are conducted in a zero-shot and task-specific transfer setting without any task-specific fine-tuning, this inevitably introduces fluctuations in performance. This behavior is consistent with the inherent challenges of cross-domain generalization and semantic transfer in open-world environments.
>
> * **Net Positive Gain**: Crucially, the overall intrusion evaluation metrics consistently show a net positive gain across the normal or adverse dataset. This proves that the methods can effectively improve the intrusion detection performance and provide a stable, generalizable solution in some of the challenging scenarios.
>
> * **Stability Demonstration**: The best evidence for the stability of our gain is seen in the consistent performance across multiple datasets, where OVIDNet consistently outperforms the baseline. This demonstrates that the improvements are not accidental, but rather indicative of a fundamentally more robust representation learned by incorporating DMG and MDNM.

---

> ### Author Response · Authors · 2025-11-26
>
> **W5: The absolute performance remains low (32.79 percent intrusion accuracy), and the Rider class fails completely (0.0 percent) due to confusion with Person and class imbalance. The paper does not address this failure.**
>
> **A5**: Thanks!  Regarding the question, we would like to explain it from the following two aspects for you:
>
> **(1) Sub-weakness1: The absolute performance remains low**
>
> **(i) Novel task and first try**: In our paper, we introduce for the first time an open-vocabulary intrusion detection task to address the needs of intrusion detection in open-world environments. To the best of our knowledge, the task of dynamic-view Open Vocabulary Intrusion Detection is proposed for the first time. This is the first multi-modal try in the vision-based intrusion task. We also propose and design corresponding models and strategies to better accomplish the OVID task. Therefore, this is an entirely new domain and the first attempt, so we consider this to be normal. Besides, we also tested some existing models, e.g., OpenSeed model, and found that current models do not perform well on our OVID task either.
>
> **(ii) Transfer performance**: In our experiments, we adopt the two common ways,  zero-shot and task-specific transfer settings. Our model was not obtained through fully supervised training. This was primarily to evaluate its performance in open-world settings.
>
> **(iii) Intrusion paradigm innovation**: Note that we are the first to pioneer open-world intrusion detection research based on open vocabulary. Our work represents a paradigm-shifting contribution. Simultaneously, the accuracy indicates that some work remains to be done and refined in this field and direction, which will be our future focus.
>
> **(2) Sub-weakness2: The Rider class fails completely (0.0 percent) due to confusion with Person and class imbalance.**
>
> To compensate for this gap and enhance the practicality of our model, we design a simple yet effective reasoning enhancement strategy, as follows:
>
> **(i) Geometric Constraint Reclassification Strategy (GCRS)**: The GCRS strategy is implemented within the structured post-processing phase of the object detection pipeline, specifically designed to mitigate local class ambiguity arising from the model's decomposition of compound semantic entities like the Rider, e.g., category similarity. The ambiguity occurs when the model predicts the person on the vehicle as a standalone Subject Box $B_P$ (Person), rather than the intended composite Rider class. The principle of the GCRS strategy relies on utilizing the spatial topological constraints between predicted bounding boxes. By calculating the IoU between the detected $B_P$ and any potential Accessory Box $B_V$ (Bicycle/Motorcycle), we quantify their degree of spatial coupling. If this coupling exceeds a predefined High-Coupling Threshold $\tau$, it constitutes a strong geometric constraint, necessitating a mandatory class reassignment to enforce semantic consistency. The core correction criterion is formally expressed as a conditional reclassification operation:
>
> $$\text{If } \text{Class}(B_P) = \text{'Person'} \text{ and } \left( \exists B_V \in \{(\text{'Bicycle'}, \text{'Motorcycle'})\} \text{ s.t. } \text{IoU}(B_P, B_V) > \tau \right), \text{Then } \text{Class}(B_P) \leftarrow \text{'Rider'}$$
>
> where $\tau$ is a threshold. The IoU metric used to quantify the overlap is defined as:
>
> $$\text{IoU}(B_P, B_V) = \frac{\text{Area}(B_P \cap B_V)}{\text{Area}(B_P \cup B_V)}$$
>
> GCRS strategy acts as a domain-knowledge-driven geometric filter, significantly enhancing the model's accuracy on these challenging composite instances by prioritizing geometric evidence over primary model scores.
>
> **(ii) Experiment results**: Finally, we conduct several experiments to verify the effectiveness of the proposed GCRS strategy, as shown below. We can find that when applying the proposed strategy, the performance of the rider category improved from 0 to 23.56. Additionally, intrusion detection performance has improved, i.e., 32.79→33.55. In future research, we will explore more effective strategies to continue improving performance.
>
> | Performance      | Rider_Y |Rider_N |Rider |Acc
> | ----------- | ----------- |----------- |----------- |----------- |
> | OVIDNet (ours)      | 0      | 0 | 0 | 32.79
> | + GCRS   | 11.95      | 31.52 | 23.56 | **33.55**

---

> ### Author Response · Authors · 2025-11-26
>
> **W6: Comparisons with other strong open-vocabulary detection systems, such as YOLO-World and Grounding DINO, are missing, which weakens the SOTA claim.**
>
> **A6**: Thanks for your good suggestions!
>
> * **Some investigation and findings**: We conduct a thorough investigation of these two models, i.e., YOLO-World and Grounding DINO, and find that they primarily focus on the open-vocabulary detection domain. However, our task requires not only detection sub-tasks but also segmentation sub-tasks and intrusion detection sub-tasks. Therefore, open-vocabulary detection models alone cannot fulfill our intrusion detection requirements. Besides, we also find that some models support the open-vocabulary instance segmentation. However, these models also cannot meet the needs of the OVID task. Let us make a comparison.
> | Name      | Instance segmentation | Panoptic segmentation
> | ----------- | ----------- | ----------- |
> | Does it segment stuff (road/other)?     |  $\times$       | $\checkmark$
> | Are output pixels fully covered?  |  $\times$       | $\checkmark$
> | Can meet the requirements of the OVID task?   |  $\times$       | $\checkmark$
>
> * **Extra experiments**: Nevertheless, to validate the effectiveness of our proposed model, we conducted some experiments on the YOLO-world and GroundingDINO models. Specifically, we follow previous intrusion detection work [5] [6] and adopt a combined model approach for testing, i.e., open-vocabulary detection (OVD) + open-vocabulary segmentation (OVS). Here, open-vocabulary detection models contain YOLO-world. Besides, the open-vocabulary segmentation model adopts the Clipseg to complete the experiments. Note that for the latter (open-vocabulary segmentation model), we initially intended to adopt the SAM model. However, we find that the original SAM requires point or bounding box information for object segmentation. Therefore, in these experiments, we employed the Clipseg model for the open-vocabulary segmentation subtask. The specific experiment results are as follows:
> | Model | Type | Domain | Acc | Parameter |
> | :---: | :---: | :---: | :---: | :---: |
> | YOLO-World-S [7] +Clipseg [8] | OVD+OVS | Normal | 22.64 | 163.75M |
> | | | Foggy | 17.75 | |
> | YOLO-World-L [7] +Clipseg [8]  | OVD+OVS | Normal | 30.08 | 198.75M |
> | | | Foggy | 24.34 | |
> | **OVIDNet (Ours)** | End-to-End | Normal | **32.79** | **120.32M** |
> | | | Foggy | **27.83** | |
>
> **W7: The dataset labeling relies on an automatic 20-pixel overlap rule with limited manual verification. There is no inter-annotator agreement, threshold sensitivity analysis, or bias auditing.**
>
> **A7**: Thanks! Let us explain to you in detail.
>
> * **Default settings**. In the field of intrusion detection, the widely accepted pixel threshold is 20. This is primarily because early definitions [5] and studies on intrusion detection consistently set the original pixel threshold to 20 [6][9]. This value is not empirical but derived from experimentation.  Additionally, we conduct manual verification, as detailed in Appendix A.1 (step 8).
>
> * **Additional experiments**:  We also conduct some sensitivity experiments to the threshold t in a more adverse environment dataset. For detailed information about the dataset, please refer to the answer (A7) for Weakness 7. Here, we set the threshold t as 10, 15, 20, 30, respectively, and the results are shown below. We can find that our proposed strategy and framework are insensitive to pixel thresholds. Under the different pixel values, our performance consistently outperforms the baseline and achieves the best results. Besides, as different strategies are added, intrusion performance continues to improve. We can also see that our best performance can still be improved, e.g., when the pixel is 30. The primary reason is that as the pixels increases, the model's sensitivity to intrusion behavior decreases while its sensitivity to non-intrusion behavior increases. According to the computational method employed [5][6], this will lead to an overall increase in the model's sensitivity to intrusions. Therefore, these sensitivity experiments further verify the robustness and efficacy of the proposed strategy and framework across varying environments and settings.
> | Method     |  | t=10  | |  | t=15  | | | t=20  | |  | t=30  | |
> | ----------- | ----------- | ----------- | ----------- | ----------- | ----------- | ----------- | ----------- | ----------- | ----------- | ----------- | ----------- | ----------- |
> |  - |AccY|AccN|Acc|AccY|AccN |Acc |AccY | AccN |Acc | AccY | AccN |Acc |
> |  Baseline |22.32 |15.74 |18.31 |21.64 | 16.55 | 18.54 |20.99 | 17.22 |18.69 |19.93 | 18.45 |19.03 |
> |  +DMG | 21.86 | 17.30 | 19.09 |20.83 | 18.27 | 19.27 | 20.16 | 19.27 | 19.62 |19.15 | 20.75 |20.13 |
> |  +MDNM | 21.81 | 18.17 |19.60 |20.96 | 19.34 |19.97 |20.26 | 20.38 | 20.33 |19.08 | 22.06 |20.89 |
> |  +DMG+MDNM | 28.30 | 19.01 | 22.64 |26.74 | 20.46 |22.92 |**25.79** | **21.66** |**23.27**|24.45| 23.59 |23.93 |

---

> ### Author Response · Authors · 2025-11-26
>
> **-----Questions section response-----**
>
> **Q1: Why is only selectively synthetic fog evaluated?,......, Exploring how AoI–box consistency behaves under viewpoint changes and visual artifacts could provide valuable insights into the model’s robustness.**
>
> **A1**: Good question! To consider more distribution shift types and enhance the diversity of intrusion scene in open-world deployment, we created a new intrusion detection dataset for the OVID task, namely Cityintrusion-OpenV-BDD. The new dataset is built based on the BDD-100K datasets [3]. We clean the original dataset based on the proposed OVID task features. The detailed method can refer to Appendix A.1 in our paper. Our new datasets contain rich intrusion scene types, e.g., multiple different weather (Clear, Cloudy, Rainy, Foggy, Night), different geographic environments (City, Highway, Suburban/Rural), different period of time (Daytime, Dusk, Night), and Different transportation environments (Heavy Traffic, Empty Road). These new domains can meet the experiment's requirements in different distribution shifts. Finally, our datasets contain 1482 training data and 449 evaluation data. We evaluate the performance of our model on these datasets, as shown in the table below. We can find that, in different domain shifts, our strategies still present performance improvements. Compared with the baseline model, our model can surpass it by **4.58%**, which verifies the strong robustness of our model and the effectiveness of the proposed strategies.
> | Baseline | DMG | MDNM | AccY (%) | AccN (%) | Acc (%) | Gain |
> | :---: | :---: | :---: | :---: | :---: | :---: | :---: |
> | $\checkmark$ |$\times$ | $\times$ | 20.99 | 17.22 | 18.69 |- |
> | $\checkmark$ | $\checkmark$ | $\times$ | 20.16 | 19.27 | 19.62 | +0.93 |
> | $\checkmark$ | $\times$ | $\checkmark$ | 20.26 | 20.38 | 20.33 | +1.64 |
> | $\checkmark$ | $\checkmark$ | $\checkmark$ | 25.79 | 21.66 | **23.27** | +4.58 |
>
> **Q2: What happens if the models are trained at higher resolution (1200) and for longer schedules? Do the reported gains hold?**
>
> **A2**: Thank you! For the detailed gains change at higher resolution (1200) and for longer schedules, please refer to the answer for Weakness 1. Thanks again!.
>
> **Reference**
>
> [1] Gandikota K V, Chandramouli P. Text-guided explorable image super-resolution[C]//Proceedings of the IEEE/CVF Conference on Computer Vision and Pattern Recognition. 2024: 25900-25911.
>
> [2] Korkmaz C, Tekalp A M, Dogan Z. Training generative image super-resolution models by wavelet-domain losses enables better control of artifacts[C]//Proceedings of the IEEE/CVF Conference on Computer Vision and Pattern Recognition. 2024: 5926-5936.
>
> [3] Yu F, Chen H, Wang X, et al. Bdd100k: A diverse driving dataset for heterogeneous multitask learning[C]//Proceedings of the IEEE/CVF conference on computer vision and pattern recognition. 2020: 2636-2645.
>
> [4] Chapelle O, Weston J, Bottou L, et al. Vicinal risk minimization[J]. Advances in neural information processing systems, 2000, 13.
>
> [5] Sun J, Chen J, Chen T, et al. PIDNet: An efficient network for dynamic pedestrian intrusion detection[C]//Proceedings of the 28th ACM International Conference on Multimedia. 2020: 718-726.
>
> [6] Han F, Ye P, Li K, et al. Mf-id: a benchmark and approach for multi-category fine-grained intrusion detection[J]. IEEE Transactions on Automation Science and Engineering, 2024, 22: 3582-3597.
>
> [7] Cheng T, Song L, Ge Y, et al. Yolo-world: Real-time open-vocabulary object detection[C]//Proceedings of the IEEE/CVF conference on computer vision and pattern recognition. 2024: 16901-16911.
>
> [8] Lüddecke T, Ecker A. Image segmentation using text and image prompts[C]//Proceedings of the IEEE/CVF conference on computer vision and pattern recognition. 2022: 7086-7096.
>
> [9] Shi Z, He S, Sun J, et al. An efficient multi-task network for pedestrian intrusion detection[J]. IEEE transactions on intelligent vehicles, 2022, 8(1): 649-660.

---

### Official Review · Reviewer_7Wz9 · 2025-11-03

**Soundness:** 3
**Presentation:** 2
**Contribution:** 2
**Rating:** 4
**Confidence:** 3

**Summary:**

This paper proposes the first Open-Vocabulary Intrusion Detection task, builds the Cityintrusion-OpenV dataset (with 8 intrusion categories), and designs the end-to-end OVIDNet framework. Then they provide OVIDNet, their end-to-end framework that pairs CLIP for text and a tiny Swin Transformer for images, checking overlaps between bounding boxes and AOIs to judge intrusions.

**Strengths:**

Open-vocab was a gap in intrusion detection, and the Cityintrusion-OpenV dataset provides new annotations for this task.

**Weaknesses:**

1. The core logic for “intrusion judgment” feels rather vague. As it stands, the method appears to detect any object that appears on the road, rather than accurately identifying what constitutes an intrusion. Although the paper mentions using the overlap between bounding boxes and the AoI to determine intrusions, it never clearly defines what constitutes an intrusion in different scenarios. For instance, are regular vehicles considered intruders? What about pedestrians or bicycles? Are there specific time periods or restricted zones where entering the AoI becomes an intrusion? Without these clarifications, the task essentially boils down to “object detection plus overlap checking,” rather than genuine intrusion detection, which should focus on identifying unauthorized entries. In fact, if the goal is simply to detect everything on the road, this can be easily achieved using an MLLM or by combining a segmentation model with an open-vocabulary detector.
2. Two core improvements lack mechanism proof. For the Multi-Distributed Noise Mixing strategy, the paper only states it mixes three types of noise and adjusts ratios dynamically, but fails to explain why this specific noise combination enhances unknown category location information or how dynamic ratios avoid damaging normal features. For the Dynamic Memory-Gated Module, it only describes the structural process (global average pooling → memory retrieval → feature fusion) without clarifying why the query vector matches memory units effectively.
3. The method performs poorly on basic categories like “Person” and “Rider,” weakening its practicality for common intrusion scenarios.
4. The figures are too small to be visible clearly, hindering verification of visualization results.

**Questions:**

- When the paper mentions “multi-domain,” is it only normal and foggy weather? Most tests switch between those two, but there’s no word on other domains like rainy or night.
- Line 168 has a typo—“detention” should be “detection.”

---

> ### Author Response · Authors · 2025-11-26
> **The Detailed response for Reviewer 7Wz9**
>
> **Rebuttal Summary**: We sincerely thank the reviewer for their valuable efforts and constructive feedback. We appreciate the reviewer' recognition of our work's contributions, including: (1) the **first** Open-Vocabulary Intrusion Detection task, (2) **builds** the Cityintrusion-OpenV dataset (with 8 intrusion categories), (3) **designs** an end-to-end framework, (4) **fills** the gap in intrusion detection.
>
> We provided detailed point-by-point responses addressing your concerns below. If you have any further questions, we would be glad to discuss them. Thank you very much.
>
> **-----Weaknesses section response-----**
>
> **W1: The core logic for “intrusion judgment” feels rather vague. As it stands,......,this can be easily achieved using an MLLM or by combining a segmentation model with an open-vocabulary detector.**
>
> **A1:** Thanks! Let us provide a detailed explanation for you.
>
> * **Intrusion detection definition**: In the previous intrusion detection works, the “intrusion detection” is defined as judging whether a possible object (e.g., person) exists in the restricted area-of-interest (AoI) [1]. Based on this definition, this work proposes a novel framework, i.e., PIDNet. This framework is composed of three modules: detection, segmentation, and intrusion detection. Consequently, subsequent works have largely followed this intrusion detection paradigm, e.g., cross-PIDNet [2], MF-ID [3], and Ada-iD [4]. Inspired by these prior works, our approach also employs three sub-tasks: end-to-end open-vocabulary detection, segmentation, and intrusion judgment.
>
> * **Combined model exploration**: We follow previous intrusion detection work [1] [3] and adopt a combined model approach for testing, i.e., open-vocabulary detection (OVD) + open-vocabulary segmentation (OVS). Here, open-vocabulary detection models contain YOLO-world. Besides, the open-vocabulary segmentation model adopts the Clipseg to complete the experiments. We can find that compared with the combined model (YOLO-World-L +Clipseg), our model can surpass it by 2.71% (normal domain) and 3.49% (foggy domain). Besides, our model has fewer parameters. These performance gains demonstrate the effectiveness of our model.
> | Model | Type | Domain | Acc | Parameter |
> | :---: | :---: | :---: | :---: | :---: |
> | YOLO-World-S [5] +Clipseg [6] | OVD+OVS | Normal | 22.64 | 163.75M |
> | | | Foggy | 17.75 | |
> | YOLO-World-L [5] +Clipseg [6]  | OVD+OVS | Normal | 30.08 | 198.75M |
> | | | Foggy | 24.34 | |
> | **OVIDNet (Ours)** | End-to-End | Normal | **32.79** | **120.32M** |
> | | | Foggy | **27.83** | |
>
> * **Novel task and strategy**: Our contribution is not redefining intrusion semantics but proposing a novel task. Based on the intrusion detection paradigm and definition, in our paper, we propose a novel open-vocabulary intrusion detection task to address the limitations of pre-defined classes. To the best of our knowledge, the OVID task is proposed for the first time. And it is also the first exploration and attempt in the multimodal domain. Additionally, we also provide corresponding datasets and benchmarks for the novel task. To accomplish the above OVID task, we have also designed a framework and two effective methods for this task. The designed framework can accomplish the proposed OVID task. Meanwhile, the two effective strategies can improve the model's generalization performance in real-world scenarios. Extensive experiments and comparisons have validated the effectiveness of our approach.
>
> * **Further exploration:** Regarding your proposal to leverage MLLM for intrusion detection tasks, we consider it an excellent suggestion. We will explore MLLM-based intrusion detection paradigms in our future research and strive to enhance the performance of our intrusion detection tasks.

---

> ### Author Response · Authors · 2025-11-26
>
> **W2: Two core improvements lack mechanism proof. For the Multi-Distributed Noise Mixing strategy, the paper only states it mixes three types of noise and adjusts ratios dynamically, but fails to explain why this specific noise combination enhances unknown category location information or how dynamic ratios avoid damaging normal features. For the Dynamic Memory-Gated Module, it only describes the structural process (global average pooling → memory retrieval → feature fusion) without clarifying why the query vector matches memory units effectively.**
>
> **A2**: Thanks! Regarding the question, we would like to explain it from the following two aspects for you:
>
> **(1) Sub-weakness1: Mechanism proof for Multi-Distributed Noise Mixing strategy**
>
> To clarify why the proposed noise mixture improves generalization toward unknown categories, we provide a theoretical proof based on Vicinal Risk Minimization (VRM). In our paper, our perturbation can be written as:
>
> $B_{f}=\mathcal{C}\left [ B_{e}+Z\odot \bigtriangleup \odot \Theta ,\textbf{ 0}, \textbf{ 1} \right ],$
>
> where $Z=\alpha \cdot \textbf{N}{u}+\beta  \cdot \textbf{N}{g} +\gamma \cdot \textbf{N}_{t} $.  And
> $\textbf{N}_u $,
> $\textbf{N}_g $,
> $\textbf{N}_t $ denote noise sampled from Uniform, Gaussian, and Laplace distributions, respectively. Since $Z$ is a combination of three independent noise sources, its distribution can be expressed as:
>
> $p_Z(z) = \alpha p_u(z) + \beta p_g(z) + \gamma p_t(z),$
>
> which is a mixed distribution containing (1) fine-grained bounded perturbations (U), (2) moderate Gaussian variations (Gaussian), (3) heavy-tailed structural deviations (Laplace). This directly yields a mixed vicinal distribution in the Vicinal Risk Minimization (VRM) framework. Following Chapelle et al. [7], the vicinal risk can be written as:
>
> $\hat{R} _ {\text{VRM}}(f) = \mathbb{E} _ {(x,y) \sim \mathcal{D}} \mathbb{E} _ {z \sim \mathcal{ p _ Z }} \ell(f(x+z), y).$
>
> Because $\mathcal{ p_Z }$ is a mixture, VRM can be expressed as:
>
> $\hat{R}\_{\text{M-VRM}}(f) = \alpha \hat{R}_u(f) + \beta \hat{R}_g(f) + \gamma \hat{R}_t(f),$
>
> where $\hat{R} _ {u}(f) = \mathbb{E} _ {z \sim p _ u} \ell (f(x+z), y), \hat{R} _ {g}(f) = \mathbb{E} _ {z \sim p_g} \ell (f(x+z), y), \hat{R} _ {t}(f) = \mathbb{E} _ {z \sim p_t} \ell (f(x+z), y).$ Thus, instead of training against a single perturbation model, our model effectively optimizes risk under three complementary vicinal neighborhoods simultaneously.
>
> Let the real open-world perturbation distribution be $q(z)$. If $q$ and $p_Z$ share support and $q \ll p_Z$, then the Radon-Nikodym ratio can be expressed as:
>
> $$K := \sup_z \frac{q(z)}{p_Z(z)},$$
>
> and the $K$ is finite, giving:
>
> $$q(z) \le K p_Z(z).$$
>
> Because $p_Z(z) = \alpha p_u(z) + \beta p_g(z) + \gamma p_t(z)$ covers bounded signals (U), smooth variations (Gaussian), and heavy-tailed structural changes (Laplace). It has wider support and a smaller worst-case ratio $𝐾$ than any single distribution. Therefore, the true risk is upper-bounded by:
>
> $ R_q(f) =  \mathbb{E} _ {z \sim q} \ell (f(x+z), y) \le K \hat{R} _ {\text{M-VRM}}(f). $
>
> A smaller constant $𝐾$ implies a tighter bound. Hence, our model can better generalize to unknown object shapes, sizes, and distributions, and improve localization for novel or rare categories. Because their perturbations are more likely to fall inside the large support of the mixture $p_Z$.

---

> ### Author Response · Authors · 2025-11-26
>
> **(2) Sub-weakness2: Mechanism Explanation for Dynamic Memory-Gated Module**
>
> **Mechanism analysis**: We appreciate the reviewer's detailed attention to the underlying mechanism. The effectiveness of the query vector $\mathbf{Q}$ in matching memory units is proven through the implementation of a Content-Addressable Memory realized via Scaled Dot-Product Attention, which provides a formally defined, evidence-based retrieval process.
>
> The query vector $\mathbf{Q} \in \mathbb{R}^{B \times C}$ is obtained via Global Average Pooling (GAP) on the input features $\mathbf{X}$. This operation enforces $\mathbf{Q}$ to be the spatial average of the feature channels, $\mathbf{Q} = \text{GAP}(\mathbf{X})$. Mathematically, $\mathbf{Q}$ represents the global semantic descriptor of the input scene, providing the necessary high-level context for retrieval.
>
> The core mechanism for effective matching is defined by Equation (4):
>
> $$\mathbf{O}_m = \text{softmax}\left(\frac{\mathbf{Q}\mathbf{M}_K^T}{\sqrt{d}}\right) \mathbf{M}_V$$
>
> The term $\mathbf{A} = \mathbf{Q}\mathbf{M}\_K^T$ computes the unnormalized attention scores, where each element $A_{b, m}$ in $\mathbf{A}$ is the dot product between the batch-wise query $\mathbf{Q}\_b$ and the $m$-th memory key $\mathbf{M}\_{K, m}$. Since $\mathbf{M}\_K$ are learned, fixed representations encoding various long-term scene dependencies, the dot product acts as a measure of cosine similarity (up to normalization) between the current scene's global context $\mathbf{Q}_b$ and the stored memory key $\mathbf{M}\_{K, m}$. The $\text{softmax}(\cdot)$ operation then transforms these similarity scores into a set of dynamic attention weights for retrieval. This mechanism rigorously guarantees that a memory unit $\mathbf{M}\_{V, m}$ is weighted proportionally to its semantic relevance to the current input $\mathbf{X}$.
>
> Finally, the dynamic weights $\mathbf{W}$ from the Dynamic Gated Module, defined as $\mathbf{W} = \sigma(\mathbf{W}_2\text{ReLU}(\mathbf{W}_1\mathbf{Q}))$, act as an additional gate on the input features $\mathbf{X}$ before fusion with the memory output $\mathbf{O}_m$. This dual-branch architecture ensures the output feature $\mathbf{X}_f$ (Eq. 5) is controlled by both the global context ($\mathbf{Q}$ influences $\mathbf{W}$ and $\mathbf{O}_m$) and the memory bank ($\mathbf{M}_K, \mathbf{M}_V$), providing a robust and provable way to fuse dynamic adaptation and long-term dependency retrieval.
>
> **Empirical Demonstration of Significance**: We demonstrate the significance of the DMG module in our ablation study (Table 5). Specifically, the DMG module is vital for improving performance on the challenging Intrusion Judgment metric. Removing the DMG leads to a substantial decrease of 1.36% in the final detection and judgment accuracy.
>
> | Baseline       | Memeroy units |  Acc(%) |
> | ----------- | ----------- |----------- |
> | √      | ×      | 29.36 |
> | √    | √ (M=30) | 30.31 |
> | √    | √ (M=40) | **30.72** |
> | √    | √ (M=50) | 28.42 |
>
> **Validation on Adverse Environment Datasets**:  To verify the effectiveness of the proposed DMG strategy, we also create a new intrusion detection dataset for the OVID task, namely Cityintrusion-OpenV-BDD. The new dataset is built based on the BDD-100K datasets. Our new datasets contain rich intrusion scene types, e.g., multiple different weather (Clear, Cloudy, Rainy, Foggy, Night), different geographic environments (City, Highway, Suburban/Rural), different period of time (Daytime, Dusk, Night), and Different transportation environments (Heavy Traffic, Empty Road). We can find that, in different domain shifts, compared with the baseline model, our model can surpass it by 0.93%. These performance improvements denote that our DMG strategy is effective, especially in facing and dealing with novel and adverse intrusion scenarios.
>
> | Baseline       | Memeroy units |  Acc(%) |
> | ----------- | ----------- |----------- |
> | √      | ×      | 18.69 |
> | √      | √ (M=40)      | **19.62** |

---

> ### Author Response · Authors · 2025-11-26
>
> **W3: The method performs poorly on basic categories like “Person” and “Rider,” weakening its practicality for common intrusion scenarios.**
>
> **A3**: Good question and suggestions! To compensate for this gap and enhance the practicality of our model, we design a simple yet effective reasoning enhancement strategy, as follows:
>
> **(1) Geometric Constraint Reclassification Strategy (GCRS)**: The GCRS strategy is implemented within the structured post-processing phase of the object detection pipeline, specifically designed to mitigate local class ambiguity arising from the model's decomposition of compound semantic entities like the Rider, e.g., category similarity. The ambiguity occurs when the model predicts the person on the vehicle as a standalone Subject Box $B_P$ (Person), rather than the intended composite Rider class. The principle of the GCRS strategy relies on utilizing the spatial topological constraints between predicted bounding boxes. By calculating the IoU between the detected $B_P$ and any potential Accessory Box $B_V$ (Bicycle/Motorcycle), we quantify their degree of spatial coupling. If this coupling exceeds a predefined High-Coupling Threshold $\tau$, it constitutes a strong geometric constraint, necessitating a mandatory class reassignment to enforce semantic consistency. The core correction criterion is formally expressed as a conditional reclassification operation:
>
> $$\text{If } \text{Class}(B_P) = \text{'Person'} \text{ and } \left( \exists B_V \in \{(\text{'Bicycle'}, \text{'Motorcycle'})\} \text{ s.t. } \text{IoU}(B_P, B_V) > \tau \right), \text{Then } \text{Class}(B_P) \leftarrow \text{'Rider'}$$
>
> where $\tau$ is a threshold. The IoU metric used to quantify the overlap is defined as:
>
> $$\text{IoU}(B_P, B_V) = \frac{\text{Area}(B_P \cap B_V)}{\text{Area}(B_P \cup B_V)}$$
>
> GCRS strategy acts as a domain-knowledge-driven geometric filter, significantly enhancing the model's accuracy on these challenging composite instances by prioritizing geometric evidence over primary model scores.
>
> **(2) Experiment results**: Finally, we conduct several experiments to verify the effectiveness of the proposed GCRS strategy, as shown below. We can find that when applying the proposed strategy, the performance of the rider category improved from 0 to 23.56. Additionally, intrusion detection performance has improved, i.e., 32.79→33.55. In future research, we will explore more effective strategies to continue improving performance.
>
> | Performance      | Rider_Y |Rider_N |Rider |Acc
> | ----------- | ----------- |----------- |----------- |----------- |
> | OVIDNet (ours)      | 0      | 0 | 0 | 32.79
> | + GCRS   | 11.95      | 31.52 | 23.56 | **33.55**
>
> **W4: The figures are too small to be visible clearly, hindering verification of visualization results.**
>
> **A4**: Thanks! We will enlarge the figures in the revised version to ensure clear visibility and better verification of the visualization results.

---

> ### Author Response · Authors · 2025-11-26
>
> **-----Questions section response-----**
>
> **Q1: When the paper mentions “multi-domain,” is it only normal and foggy weather? Most tests switch between those two, but there’s no word on other domains like rainy or night.**
>
> **A1**: Good question! To consider more distribution shift types and enhance the diversity of intrusion scenes in open-world deployment, we created a new intrusion detection dataset for the OVID task, namely Cityintrusion-OpenV-BDD. The new dataset is built based on the BDD-100K datasets [8]. We clean the original dataset based on the proposed OVID task features. The detailed method can refer to Appendix A.1 in our paper. Our new datasets contain rich intrusion scene types, e.g., multiple different weather (Clear, Cloudy, Rainy, Foggy, Night), different geographic environments (City, Highway, Suburban/Rural), different period of time (Daytime, Dusk, Night), and Different transportation environments (Heavy Traffic, Empty Road). These new domains can meet the experiment's requirements in different distribution shifts. Finally, our datasets contain 1482 training data and 449 evaluation data. We evaluate the performance of our model on these datasets, as shown in the table below.
>
> We can find that, in different domain shifts, our strategies still present promising performance improvements. Compared with the baseline model, our model can surpass it by **4.58%**, which verifies the strong robustness of our model and the effectiveness of the proposed strategies.
>
> | Baseline | DMG | MDNM | AccY (%) | AccN (%) | Acc (%) | Gain |
> | :---: | :---: | :---: | :---: | :---: | :---: | :---: |
> | $\checkmark$ |$\times$ | $\times$ | 20.99 | 17.22 | 18.69 |- |
> | $\checkmark$ | $\checkmark$ | $\times$ | 20.16 | 19.27 | 19.62 | +0.93 |
> | $\checkmark$ | $\times$ | $\checkmark$ | 20.26 | 20.38 | 20.33 | +1.64 |
> | $\checkmark$ | $\checkmark$ | $\checkmark$ | 25.79 | 21.66 | **23.27** | +4.58 |
>
>
> **Q2: Line 168 has a typo—“detention” should be “detection.**
>
> **A2**: Thanks! We will review our paper carefully and make minor revisions to address the issues.
>
> **Reference**
>
> [1] Sun J, Chen J, Chen T, et al. PIDNet: An efficient network for dynamic pedestrian intrusion detection[C]//Proceedings of the 28th ACM International Conference on Multimedia. 2020: 718-726.
>
> [2] Shi Z, He S, Sun J, et al. An efficient multi-task network for pedestrian intrusion detection[J]. IEEE transactions on intelligent vehicles, 2022, 8(1): 649-660.
>
> [3] Han F, Ye P, Li K, et al. Mf-id: a benchmark and approach for multi-category fine-grained intrusion detection[J]. IEEE Transactions on Automation Science and Engineering, 2024, 22: 3582-3597.
>
> [4] Han F, Ye P, Duan S, et al. Ada-iD: Active Domain Adaptation for Intrusion Detection[C]//Proceedings of the 32nd ACM International Conference on Multimedia. 2024: 7404-7413.
>
> [5] Cheng T, Song L, Ge Y, et al. Yolo-world: Real-time open-vocabulary object detection[C]//Proceedings of the IEEE/CVF conference on computer vision and pattern recognition. 2024: 16901-16911.
>
> [6] Lüddecke T, Ecker A. Image segmentation using text and image prompts[C]//Proceedings of the IEEE/CVF conference on computer vision and pattern recognition. 2022: 7086-7096.
>
> [7] Chapelle O, Weston J, Bottou L, et al. Vicinal risk minimization[J]. Advances in neural information processing systems, 2000, 13.
>
> [8] Yu F, Chen H, Wang X, et al. Bdd100k: A diverse driving dataset for heterogeneous multitask learning[C]//Proceedings of the IEEE/CVF conference on computer vision and pattern recognition. 2020: 2636-2645.

---

### Official Review · Reviewer_v3N4 · 2025-11-07

**Soundness:** 2
**Presentation:** 3
**Contribution:** 2
**Rating:** 4
**Confidence:** 4

**Summary:**

This paper establishes a new task of dynamic-view Open Vocabulary Intrusion Detection, including a correlated dataset Cityintrusion-OpenV. An end-to-end model, OVIDNet, is designed as a baseline for the proposed new benchmark. Two strategies are proposed to improve the generalization and performance of OVIDet.

**Strengths:**

1. The proposed dataset extends the future research area of vision intrusion detection.
2. The overall method is easy to follow.
3. Comprehensive experiments and comparisons are conducted to verify the effectiveness of the proposed framework and methods.

**Weaknesses:**

1. The designed Multi-Distributed Noise Mixing Strategy is an incremental enhancement of the noise generation in OpenSeeD.
2. While the Dynamic Memory-Gated module is designed to capture the contextual information under complex scenarios, no discussion or analysis demonstrates whether this challenge is inherently significant in OVID. It is also unclear how this module functions in the overall model.
3. The open-vacabulairy capability relies on existing OVD and OVS practices. The overall method lacks in-depth insight into tackling the proposed OVID task.
4. While the established challenge is an open-vocabulary problem, the experiments are conducted under zero-shot and task-specific transfer settings (lines 306-307).
Other typo issues, including but not limited to
1. Line 010, using e.g. and etc. in the same sentence.
2. Line 173, "(D_T^d, D_V^s)" . I believe this should be "(D_T^d, D_T^s)"
3. Line 315, "duo to"

**Questions:**

1. What does "model singularity" in line 088 mean?
2. What does \textbf{B} denote in Table 4 and Table 5?
3. Is it sufficient to solve the OVID problem by adding the intrusion judgment capability to an open-vocabulary detection and segmentation model? In this work, the implementation of intrusion judgment is simple and learning-free, which suggests that simply combining the two should work for the OVID task. The authors should make an in-depth discussion or analysis to reveal the inherent significance of the proposed OVID task.

---

> ### Author Response · Authors · 2025-11-26
> **The Detailed response for Reviewer v3N4**
>
> **Rebuttal Summary**: We sincerely thank the reviewer for their valuable efforts and constructive feedback. We appreciate the reviewer' recognition of our work's contributions, including: (1) the **contribution** of establishing a **new** dynamic-view open-vocabulary intrusion detection task, (2) the **introduction** of the CityIntrusion-OpenV dataset, (3) **extends** the future research area of vision intrusion detection, (4) the **clarity** and **accessibility** of our proposed OVIDNet framework and methodology, (5) the **comprehensiveness** of our experiments, which validate the effectiveness of the proposed strategies.
>
> We provided detailed point-by-point responses addressing your concerns below. If you have any further questions, we would be glad to discuss them. Thank you very much.
>
> *-----Weaknesses section response-----*
>
> **W1: The designed Multi-Distributed Noise Mixing Strategy is an incremental enhancement of the noise generation in OpenSeeD.**
>
> **A1:** Thanks! Let us explain in detail for you:
>
> **(i) Some Distinction**: While our Multi-Distributed Noise Mixing (MDNM) builds on the idea of noise augmentation, it is not a straightforward incremental adjustment. MDNM introduces three **conceptually distinct** elements jointly targeted to the OVID problem: (1) mixing noise distributions with different statistical properties (uniform / Gaussian / Laplace) to simulate fine-grained, global and heavy-tail perturbations; (2) a **scale-aware, area-dependent** dynamic weighting scheme that adapts noise ratios to object size (so small objects receive different perturbation profiles than large ones); (3) explicit design choices to preserve normal-object features while amplifying localization cues for unknown targets. These three ingredients together create a new augmentation mechanism tailored for the open-vocabulary intrusion detection task — a behavioral regime not addressed by prior OpenSeeD augmentations.
>
> **(ii) Mechanism Proof**: Besides, we also provide detailed mechanism proof based on Vicinal Risk Minimization (VRM) for the Multi-Distributed Noise Mixing strategy, as follows:
>
> In our paper, our perturbation can be written as:
>
> $B_{f}=\mathcal{C}\left [ B_{e}+Z\odot \bigtriangleup \odot \Theta ,\textbf{ 0}, \textbf{ 1} \right ],$
>
> where $Z=\alpha \cdot \textbf{N}{u}+\beta  \cdot \textbf{N}{g} +\gamma \cdot \textbf{N}_{t} $.  And
> $\textbf{N}_u $,
> $\textbf{N}_g $,
> $\textbf{N}_t $ denote noise sampled from Uniform, Gaussian, and Laplace distributions, respectively. Since $Z$ is a combination of three independent noise sources, its distribution can be expressed as:
>
> $p_Z(z) = \alpha p_u(z) + \beta p_g(z) + \gamma p_t(z),$
>
> which is a mixed distribution containing (1) fine-grained bounded perturbations (U), (2) moderate Gaussian variations (Gaussian), (3) heavy-tailed structural deviations (Laplace). This directly yields a mixed vicinal distribution in the Vicinal Risk Minimization (VRM) framework. Following Chapelle et al. [1], the vicinal risk can be written as:
>
> $\hat{R} _ {\text{VRM}}(f) = \mathbb{E} _ {(x,y) \sim \mathcal{D}} \mathbb{E} _ {z \sim \mathcal{ p _ Z }} \ell(f(x+z), y).$
>
> Because $\mathcal{ p_Z }$ is a mixture, VRM can be expressed as:
>
> $\hat{R}\_{\text{M-VRM}}(f) = \alpha \hat{R}_u(f) + \beta \hat{R}_g(f) + \gamma \hat{R}_t(f),$
>
> where $\hat{R} _ {u}(f) = \mathbb{E} _ {z \sim p _ u} \ell (f(x+z), y), \hat{R} _ {g}(f) = \mathbb{E} _ {z \sim p_g} \ell (f(x+z), y), \hat{R} _ {t}(f) = \mathbb{E} _ {z \sim p_t} \ell (f(x+z), y).$ Thus, instead of training against a single perturbation model, our model effectively optimizes risk under three complementary vicinal neighborhoods simultaneously.
>
> Let the real open-world perturbation distribution be $q(z)$. If $q$ and $p_Z$ share support and $q \ll p_Z$, then the Radon-Nikodym ratio can be expressed as:
>
> $$K := \sup_z \frac{q(z)}{p_Z(z)},$$
>
> and the $K$ is finite, giving:
>
> $$q(z) \le K p_Z(z).$$
>
> Because $p_Z(z) = \alpha p_u(z) + \beta p_g(z) + \gamma p_t(z)$ covers bounded signals (U), smooth variations (Gaussian), and heavy-tailed structural changes (Laplace). It has wider support and a smaller worst-case ratio $𝐾$ than any single distribution. Therefore, the true risk is upper-bounded by:
>
> $ R_q(f) =  \mathbb{E} _ {z \sim q} \ell (f(x+z), y) \le K \hat{R} _ {\text{M-VRM}}(f). $
>
> A smaller constant $𝐾$ implies a tighter bound. Hence, our model can better generalize to unknown object shapes, sizes, and distributions, and improve localization for novel or rare categories. Because their perturbations are more likely to fall inside the large support of the mixture $p_Z$.

---

> ### Author Response · Authors · 2025-11-26
>
> **W2: While the Dynamic Memory-Gated module is designed to capture the contextual information under complex scenarios, no discussion or analysis demonstrates whether this challenge is inherently significant in OVID. It is also unclear how this module functions in the overall model.**
>
> **A2**: We appreciate the reviewer's detailed attention to the underlying mechanism.
>
> **(i) Mechanism analysis**: The effectiveness of the query vector $\mathbf{Q}$ in matching memory units is proven through the implementation of a Content-Addressable Memory realized via Scaled Dot-Product Attention, which provides a formally defined, evidence-based retrieval process.
>
> The query vector $\mathbf{Q} \in \mathbb{R}^{B \times C}$ is obtained via Global Average Pooling (GAP) on the input features $\mathbf{X}$. This operation enforces $\mathbf{Q}$ to be the spatial average of the feature channels, $\mathbf{Q} = \text{GAP}(\mathbf{X})$. Mathematically, $\mathbf{Q}$ represents the global semantic descriptor of the input scene, providing the necessary high-level context for retrieval. The core mechanism for effective matching is defined by Equation (4):
>
> $$\mathbf{O}_m = \text{softmax}\left(\frac{\mathbf{Q}\mathbf{M}_K^T}{\sqrt{d}}\right) \mathbf{M}_V.$$
>
> The term $\mathbf{A} = \mathbf{Q}\mathbf{M}\_K^T$ computes the unnormalized attention scores, where each element $A_{b, m}$ in $\mathbf{A}$ is the dot product between the batch-wise query $\mathbf{Q}\_b$ and the $m$-th memory key $\mathbf{M}\_{K, m}$. Since $\mathbf{M}\_K$ are learned, fixed representations encoding various long-term scene dependencies, the dot product acts as a measure of cosine similarity (up to normalization) between the current scene's global context $\mathbf{Q}_b$ and the stored memory key $\mathbf{M}\_{K, m}$. The $\text{softmax}(\cdot)$ operation then transforms these similarity scores into a set of dynamic attention weights for retrieval. This mechanism rigorously guarantees that a memory unit $\mathbf{M}\_{V, m}$ is weighted proportionally to its semantic relevance to the current input $\mathbf{X}$.
>
> Finally, the dynamic weights $\mathbf{W}$ from the Dynamic Gated Module, defined as $\mathbf{W} = \sigma(\mathbf{W}_2\text{ReLU}(\mathbf{W}_1\mathbf{Q}))$, act as an additional gate on the input features $\mathbf{X}$ before fusion with the memory output $\mathbf{O}_m$. This dual-branch architecture ensures the output feature $\mathbf{X}_f$ (Eq. 5) is controlled by both the global context ($\mathbf{Q}$ influences $\mathbf{W}$ and $\mathbf{O}_m$) and the memory bank ($\mathbf{M}_K, \mathbf{M}_V$), providing a robust way to fuse dynamic adaptation and long-term dependency retrieval.
>
> **(ii) Empirical Demonstration of Significance**: We demonstrate the significance of the DMG module in our ablation study (Table 5). Specifically, the DMG module is vital for improving performance on the challenging Intrusion Judgment metric. Removing the DMG leads to a substantial decrease of 1.36% in the final detection and judgment accuracy.
>
> | Baseline       | Memeroy units |  Acc(%) |
> | ----------- | ----------- |----------- |
> |  $\checkmark$     |  $\times$      | 29.36 |
> |  $\checkmark$    |  $\checkmark$ (M=30) | 30.31 |
> |  $\checkmark$   |  $\checkmark$ (M=40) | **30.72** |
> |  $\checkmark$    |  $\checkmark$ (M=50) | 28.42 |
>
> **(iii) Validation on Adverse Environment Datasets**:  To verify the effectiveness of the proposed DMG strategy, we also create a new intrusion detection dataset for the OVID task, namely Cityintrusion-OpenV-BDD. The new dataset is built based on the BDD-100K datasets. Our new datasets contain rich intrusion scene types, e.g., multiple different weather (Clear, Cloudy, Rainy, Foggy, Night), different geographic environments (City, Highway, Suburban/Rural), different period of time (Daytime, Dusk, Night), and Different transportation environments (Heavy Traffic, Empty Road). We can find that, in multiple different domains, compared with the baseline model, our model can surpass it by 0.93%. These performance improvements denote that our DMG strategy is effective, especially in facing and dealing with novel and adverse intrusion scenarios.
>
> | Baseline       | Memeroy units |  Acc(%) |
> | ----------- | ----------- |----------- |
> | $\checkmark$      | $\times$      | 18.69 |
> | $\checkmark$     | $\checkmark$ (M=40)      | **19.62** |

---

> ### Author Response · Authors · 2025-11-26
>
> **W3: The open-vacabulairy capability relies on existing OVD and OVS practices. The overall method lacks in-depth insight into tackling the proposed OVID task.**
>
> **A3**: Thanks! We want to highlight and emphasize our contributions and innovations, as follows:
>
> * **Novel task and dataset.** Our main contribution lies in defining and analyzing the new OVID task, which introduces task-aware, context-dependent intrusion reasoning under open-world conditions— *a problem not addressed by prior OVD/OVS works*. To the best of our knowledge, the OVID task is proposed for the first time, and this is the first try in the vision-based intrusion field. Although we have been inspired by some excellent work from previous studies, we must acknowledge that existing models designed for OVD or OVS cannot accomplish our new OVID task. The main reason is that our OVID task is composed of three sub-tasks: detection, segmentation, and intrusion detection. Sigelie OVD or OVS can not meet the requirements of the OVID task. Besides, although the OVD+OVS combination approach is feasible, we found that it requires training both an open vocabulary detection model and an open vocabulary segmentation model, resulting in extremely high training costs. Furthermore, the combined model does not support end-to-end training and inference. Detailed comparisons can be found in Table 8 of our appendix.
>
> * **Effective design and strategy.** Besides, to accomplish the novel OVID task, we design an effective and multi-modal framework.
> In response to the characteristics of the tasks we have proposed, we have also designed corresponding strategies: (1) A Multi-Distributed Noise Mixing strategy is introduced to enhance the location information of unknown and unseen categories. (2) A Dynamic Memory-Gated module is designed to capture the contextual information under complex scenarios. These two effective strategies can improve the generalization and enhance intrusion detection performance of the framework. Comprehensive experiments and comparisons denote that these designs and strategies are feasible and innovative.
>
> **W4: While the established challenge is an open-vocabulary problem, the experiments are conducted under zero-shot and task-specific transfer settings (lines 306-307).**
>
> **A4**: Thanks! We would like to clarify and explain the relationship between open-vocabulary and these two settings (zero-shot and task-specific transfer)  for you.
>
> * **Open Vocabulary Detection Paradigm**: In open vocabulary detection tasks, zero-shot testing is a widely adopted method for evaluating model performance, having been employed in several works [2-4].  The zero-shot approach aims to verify the generalization capabilities of proposed models. These conduct experiments using diverse prompts and report the corresponding performance.
>
> * **Task-specific transfer**: Additionally, some works also propose the concept and method of “task-specific transfer” [5]. This method aims to address the challenge of adapting the proposed open-vocabulary model for transfer to specific downstream tasks.
>
> * **Our work**: In our OVID task, we also follow and employ these two paradigms for experimentation and comparison. Zero-shot is used to validate the superiority of our model's generalization capabilities, and the way of task-specific transfer is utilized to assess the intrusion performance for the proposed OVID task.
>
> **W5-W7: Line 010, using e.g. and etc. in the same sentence........Line 315, "duo to"**
>
> **A5-A7**: Thanks! We greatly appreciate your valuable suggestions. These details make a significant contribution to enhancing the quality and readability of our paper. We will review our paper carefully and make minor revisions to address these issues.

---

> ### Author Response · Authors · 2025-11-26
>
> *-----Question section response-----*
>
> **Q1: What does "model singularity" in line 088 mean?**
>
> **A1**: Thanks! The “model singularity” in line 088 denotes that the currently proposed open-vocabulary detectors can only perform detection and cannot accomplish the proposed OVID task. The main reason is that our OVID task is a multi-task, namely: simultaneously executing detection, segmentation, and intrusion detection tasks. We will clarify this point and make revisions to our paper for easier reading.
>
> **Q2: What does \textbf{B} denote in Table 4 and Table 5?**
>
> **A2**: Thanks! In Tables 4 and 5, the **B** denotes the baseline model. We will clarify this point and make minor revisions to our paper for easier reading.
>
> **Q3: Is it sufficient to solve,......,the inherent significance of the proposed OVID task.**
>
> **A3**: Good suggestion! Let us provide a detailed explanation for you.
>
> * **Combined model exploration**: We follow previous intrusion detection work [6] [7] and adopt a combined model approach for testing, i.e., open-vocabulary detection (OVD) + open-vocabulary segmentation (OVS). Here, open-vocabulary detection models contain YOLO-world [4]. Besides, the open-vocabulary segmentation model adopts the Clipseg [8] to conduct the experiments. We can find that compared with the combined model (YOLO-World-L +Clipseg), our model can surpass it by **2.71%** (normal domain) and **3.49%** (foggy domain). Besides, our model has fewer parameters. These performance gains demonstrate the effectiveness of our model.
> | Model | Type | Domain | Acc | Parameter |
> | :---: | :---: | :---: | :---: | :---: |
> | YOLO-World-S [8] +Clipseg [9] | OVD+OVS | Normal | 22.64 | 163.75M |
> | | | Foggy | 17.75 | |
> | YOLO-World-L [8] +Clipseg [9]  | OVD+OVS | Normal | 30.08 | 198.75M |
> | | | Foggy | 24.34 | |
> | **OVIDNet (Ours)** | End-to-End | Normal | **32.79** | **120.32M** |
> | | | Foggy | **27.83** | |
>
> * **Intrusion detection definition**: In the previous intrusion detection works, the “intrusion detection” is defined as judging whether a possible object (e.g., person) exists in the restricted area-of-interest (AoI) [6]. Based on this definition, this work proposes a novel framework, i.e., PIDNet. This framework is composed of three modules: detection, segmentation, and intrusion detection. Consequently, subsequent works have largely followed this intrusion detection paradigm, e.g., cross-PIDNet [9] and Ada-iD [10]. Inspired by these prior works, our approach also employs three sub-tasks: end-to-end open-vocabulary detection, segmentation, and intrusion judgment.
>
> * **Pioneering work and tasks**: In our paper, we propose a novel open-vocabulary intrusion detection task to address the limitations of pre-defined classes. To the best of our knowledge, the OVID task is proposed for the first time. And it is also the first exploration and attempt in the multimodal domain. Additionally, we also provide corresponding datasets and benchmarks for the novel task. To accomplish the above OVID task, we have also designed a framework and two effective methods for this task. The designed framework can accomplish the proposed OVID task. Meanwhile, the two effective strategies can improve the model's generalization performance in real-world scenarios. Extensive experiments and comparisons have validated the effectiveness of our strategy.

---

> ### Author Response · Authors · 2025-11-26
>
> **Reference**
>
> [1] Chapelle O, Weston J, Bottou L, et al. Vicinal risk minimization[J]. Advances in neural information processing systems, 2000, 13.
>
> [2] Liu S, Zeng Z, Ren T, et al. Grounding dino: Marrying dino with grounded pre-training for open-set object detection[C]//European conference on computer vision. Cham: Springer Nature Switzerland, 2024: 38-55.
>
> [3] Yao L, Han J, Wen Y, et al. Detclip: Dictionary-enriched visual-concept paralleled pre-training for open-world detection[J]. Advances in Neural Information Processing Systems, 2022, 35: 9125-9138.
>
> [4] Cheng T, Song L, Ge Y, et al. Yolo-world: Real-time open-vocabulary object detection[C]//Proceedings of the IEEE/CVF conference on computer vision and pattern recognition. 2024: 16901-16911.
>
> [5] Zhang H, Li F, Zou X, et al. A simple framework for open-vocabulary segmentation and detection[C]//Proceedings of the IEEE/CVF International Conference on Computer Vision. 2023: 1020-1031.
>
> [6] Sun J, Chen J, Chen T, et al. PIDNet: An efficient network for dynamic pedestrian intrusion detection[C]//Proceedings of the 28th ACM International Conference on Multimedia. 2020: 718-726.
>
> [7] Han F, Ye P, Li K, et al. Mf-id: a benchmark and approach for multi-category fine-grained intrusion detection[J]. IEEE Transactions on Automation Science and Engineering, 2024, 22: 3582-3597.
>
> [8] Lüddecke T, Ecker A. Image segmentation using text and image prompts[C]//Proceedings of the IEEE/CVF conference on computer vision and pattern recognition. 2022: 7086-7096.
>
> [9] Shi Z, He S, Sun J, et al. An efficient multi-task network for pedestrian intrusion detection[J]. IEEE transactions on intelligent vehicles, 2022, 8(1): 649-660.
>
> [10] Han F, Ye P, Duan S, et al. Ada-iD: Active Domain Adaptation for Intrusion Detection[C]//Proceedings of the 32nd ACM International Conference on Multimedia. 2024: 7404-7413.

---

### Author Response · Authors · 2025-12-04
**Summary of contributions and Rebuttal Qutcomes**

We sincerely thank all reviewers, ACs, and PCs for their invaluable efforts in the review process. Below, we summarize our core contributions and the key outcomes of the rebuttal.

**1. Core Contributions Recap**:

Our work effectively addresses a key challenge in the field of intrusion detection, i.e., their reliance on pre-defined classes limits applicability in open-world scenarios. In our paper, we introduce the Open-Vocabulary Intrusion Detection (OVID) project **for the first time** to overcome this limitation. Our key innovations and contributions are as follows:

* **Novel task and dataset**. To the best of our knowledge, the task of dynamic-view Open Vocabulary Intrusion Detection is proposed for the first time. This is the first multi-modal try in the vision-based intrusion task. A new benchmark, including a dataset called Cityintrusion-OpenV, and some strong baselines, is given for this task.

* **Effective design and strategy**. An effective, multi-modal, multi-task, and end-to-end framework, OVIDNet, is designed as a strong baseline for this new benchmark. Besides, two effective strategies are proposed to improve the generalization and performance of the framework, including the MultiDistributed Noise Mixing Strategy and the Dynamic Memory-Gated module.

* **Adequate experiments and strong results**. Comprehensive experiments and comparisons are conducted to verify the effectiveness of the proposed framework and methods. The results show that our framework not only reaches the current SOTA level but also maintains strong high practicality with task-specific transfer and zero-shot prediction abilities.

**2. Summary of Strengths**:

We sincerely thank all reviewers for their valuable efforts and constructive feedback. We appreciate the reviewer's recognition of our work's contributions, including:

* **Reviewer v3N4**: (1) the **contribution** of establishing a **new** dynamic-view open-vocabulary intrusion detection task, (2) the **introduction** of the CityIntrusion-OpenV dataset, (3) **extends** the future research area of vision intrusion detection, (4) the **clarity** and **accessibility** of our proposed OVIDNet framework and methodology, (5) the **comprehensiveness** of our experiments, which validate the effectiveness of the proposed strategies.

* **Reviewer 7Wz9**: (1) the **first** Open-Vocabulary Intrusion Detection task, (2) **builds** the Cityintrusion-OpenV dataset (with 8 intrusion categories), (3) **designs** an end-to-end framework, (4) **fills** the gap in intrusion detection.

* **Reviewer jM2n**: (1) the **first** to formally define open-vocabulary intrusion detection, (2) a **solid step** toward open-world evaluation, (3) **broad** experimental coverage and ablation results, (4) clear figures and algorithm descriptions, **sufficient** details for reproduction, and **easy** to follow, (5) **strong** potential for practical deployment. Besides, we are pleased to see that the reviewer jM2n is open to raising the score, as indicated in the comment: **"I am open to revising my score."**

* **Reviewer aUMD**: (1) **introduces** OVID, an “open-vocabulary intrusion detection” setting, (2) a **new** dataset (Cityintrusion-OpenV) built and **supports** both open-vocabulary evaluation and the intrusion decision, (3) **easy** to follow the approach, (4) **plug-and-play** memory module, (5) **well aligned** with an intrusion task.

**3. Detailed Point-by-Point Responses:**

In the rebuttal, we provided detailed, point-by-point responses to each reviewer, addressing all comments individually and thoroughly. We carefully **clarified all concerns, provided detailed mechanism proof and theoretical analysis, and conducted additional sufficient experimental results** to ensure that all questions are fully resolved. We sincerely invite the AC and the reviewers to refer to our comprehensive, line-by-line rebuttal responses below, where each issue is addressed with precise explanations and supporting analysis.

We hope these summaries make our contributions clearer and sincerely hope that the ACs and PCs will consider our work. We believe OVID makes critical contributions to the community: it provides the **first and comprehensive** benchmark, including a novel dataset, multi-modal framework, effective strategies, and strong baselines for the open-vocabulary intrusion detection task.

We would like to express our sincere gratitude once again to all the reviewers, ACs, and PCs for their time and effort in reviewing our paper.

---

### Meta-Review · Area_Chair_tY7c · 2025-12-05

**Summary:**

The reviewers generally agree that the paper introduces a new and meaningful problem formulation for open vocabulary intrusion detection and provides a valuable dataset that can stimulate future research.

At the same time, important concerns remain. Several reviewers question the technical novelty of the proposed modules. The theoretical explanations offered in the rebuttal clarify the intention but do not fully resolve whether the proposed mechanisms are necessary or superior to standard alternatives. The lack of quantitative results on real world datasets limits the strength of the claims regarding practical applicability.

Despite these limitations, the combination of a new benchmark, a clear task formulation, and a working baseline framework provides value to the community. **On balance, the submission falls into the borderline acceptance range.**

**Reviewer Concerns:**

**Concerns addressed by the rebuttal**

The authors provided higher resolution and long schedule experiments that address concerns about unfair comparisons with OpenSeeD. Multiple reviewers requested additional baselines, and the authors supplied comparisons with YOLO World and Grounding DINO combined with open vocabulary segmentation modules. Reviewer questions about the mechanism of the two proposed strategies were answered with detailed derivations and expanded explanations. Reviewer concerns about stability were addressed by reporting multi seed results that show low variance. The rebuttal also clarified text prompt formats and the handling of novel classes in the zero shot evaluation.

**Concerns that remain open**

Several reviewers question the depth of technical novelty. The rebuttal offers explanations but does not fully resolve whether the modules provide genuine architectural advances rather than incremental modifications. The intrusion decision threshold remains fixed across different image scales, and although sensitivity studies were added, a more principled formulation would be expected for a deployment oriented task. The Rider category remains weak and the improvement relies on a rule based post processing step, which raises questions about the robustness of the core model. Reviewer requests for quantitative results on real world datasets cannot be addressed in this cycle, which limits claims about practical deployment. The dataset labeling procedure and its reliance on the twenty pixel rule remain partially justified and would have benefited from further validation or calibration studies.

**Reviewer Scores:**

**Reviewer v3N4** initially assigned a score of 4. Their review acknowledges the value of the benchmark and the clarity of the task definition, but they remain concerned about the limited conceptual novelty and the incomplete justification for several design choices. The rebuttal helps clarify some details, yet it does not fundamentally resolve the reviewer’s main reservations. It is therefore reasonable to expect that the score would **remain at 4**.

**Reviewer 7Wz9** also began with a score of 4. The authors supplied additional baselines, higher resolution experiments, and extended training schedules. These additions address part of the reviewer’s concerns, although questions regarding the strength of the technical contribution and the practical robustness of the method remain. A modest improvement is possible, but given the remaining uncertainties, the most conservative expectation is that the score would remain at 4, with **a rise to 6** being possible but not guaranteed.

**Reviewer jM2n** gave a positive evaluation with an initial score of 6. Their comments suggested that they were open to raising the score if the authors demonstrated stronger empirical support and clarified the benefit of the proposed modules. The rebuttal satisfies several of these requests, including stability studies, ablations on the two strategies, and comparisons with more competitive baselines. Under these circumstances, an increase** from 6 to 8** is plausible.

**Reviewer aUMD** assigned an initial score of 6. They raised several detailed technical and empirical questions, many of which were addressed by the new results and clarifications provided in the rebuttal. Although some broader issues remain, there is a clear improvement in the completeness of the empirical evaluation. An upward adjustment is possible, but the safest expectation is that the reviewer would **keep the score at 6**.

---

### Decision · Program_Chairs · 2026-01-26

Accept (Poster)